🔓 | **Open Peer Review** | Bacteriophages | Research Article

# Disruption of bacteriophage integration site promotes rapid diversification of multicellular traits in *Bacillus subtilis*

Maja Popović,[1] Tomaž Accetto,[1] Yasmine Dergham,[2] Romain Briandet,[2] Iztok Dogša,[1] Anna Dragoš[1]

**ABSTRACT** Certain bacteria are known for their remarkable genetic and phenotypic diversity, as well as rapid morphological diversification during evolution experiments. An example is *Bacillus subtilis*, which can switch motility, biofilm, or antagonistic interaction patterns. Here, we investigated how different forms of disruption at the *spsM* locus, including SPβ integration, insertional mutagenesis (*spsM::kan*), and markerless *spsM* deletion, influence colony morphology, motility, and the emergence of spontaneous variants in *B. subtilis* natural isolates. We reassessed a previously reported biofilm defect of an *spsM::kan* mutant and found that the phenotype stemmed from an undetected secondary mutation rather than from loss of *spsM*. We observed that *spsM::kan* mutants frequently developed spontaneous mutations in key regulators of swarming motility and biofilm development. Consistently, we show that *spsM::kan* significantly elevates mutation rates, explaining why unnoticed mutations can arise rapidly during strain construction and phenotyping. In contrast, a markerless Δ*spsM* strain did not show a detectable increase in mutation rate relative to wild type, indicating that the elevated mutation rate is not attributable to loss of SpsM function. The SPβ lysogen produced far fewer visible variant morphotypes, indicating that reversible prophage integration does not lead to the same degree of diversification observed in the *spsM::kan* background. Our findings show that different modes of disrupting the *spsM* locus can alter the likelihood of selecting recurrent regulatory mutations, highlighting how local genomic context shapes phenotypic diversification. This work highlights the interplay between prophage integration, local genome architecture, and the selective pressures that influence diversification of bacterial multicellular behaviors.

**IMPORTANCE** Prophages, defined as viruses integrated into bacterial genomes, can reshape bacterial physiology and evolution. Previous studies suggested that disruption of an integration site (*spsM*) by the SPβ prophage impairs biofilm formation in *Bacillus subtilis*. Here, we show that insertion of a kanamycin resistance cassette at the native *spsM* locus (*spsM::kan*) promotes the rapid emergence of spontaneous mutations in key regulatory genes. In contrast, a markerless Δ*spsM* strain does not show a detectable increase in mutation rate, indicating that elevated mutation supply is not a general consequence of *spsM* loss. Our results indicate that different modes of *spsM* disruption have distinct consequences for phenotypic diversification. These findings help clarify earlier observations and show that phenotypic diversification depends strongly on the mode of *spsM* disruption and the genetic background. This has broader implications for how we understand the genetic basis of microbial adaptation, the genetic manipulation, and the evolutionary roles of prophages.

**KEYWORDS** colony morphology, prophage, diversification, mutation, *Bacillus subtilis*, molecular evolution, *spsM*, social traits

**Peer Reviewers** Avigdor Eldar, Tel Aviv University, Tel Aviv, Israel; Sébastien Wielgoss, Eidgenossische Technische Hochschule Zurich, Zurich, Switzerland

Address correspondence to Anna Dragoš, anna.dragos@bf.uni-lj.si.

The authors declare no conflict of interest.

See the funding table on p. 25.

B acteria adapt rapidly to new ecological conditions, a phenomenon that makes them an ideal experimental model for tracking evolution in real time and decoding molecular evolutionary patterns (1, 2). In structured environments such as solid media, multiple evolving lineages can coexist for extended periods due to spatial structure and niche separation (3, 4). In some cases, bacterial diversification becomes macroscopically visible, as outgrowths emerging from original colony outpace the ancestor in spreading across solid media (5, 6).

Certain bacterial species exhibit exceptionally high levels of genetic diversity (7), which is often reflected in striking phenotypic variation (8). A well-studied example is *Bacillus subtilis*, a soil-dwelling, spore-forming bacterium well known for its developmental plasticity, including its ability to form biofilms (9) and undergo sporulation (10). In biofilm evolution experiments, *B. subtilis* has been shown to diversify into distinct morphotypes within just a few days (11–13). This rapid diversification is also commonly observed during laboratory domestication and even through routine strain-sharing practices, often resulting in inconsistent phenotypes among strains thought to be genetically identical (14). Most reported mutations are those that are associated with an easily recognizable phenotype, such as changes in motility, changes in biofilm formation, or emergence of antagonistic interactions (15, 16). Notably, the latter two have been associated with the acquisition of new prophage elements, which are particularly prevalent in *B. subtilis* (17).

Many natural isolates of *B. subtilis* carry multiple prophages (17) with members of the *Spbetavirus* genus, particularly the SPβ phage, being among the most thoroughly characterized (18, 19). SPβ is a temperate siphovirus phage, approximately 130 kb in size (20). It was previously shown that SPβ can rapidly alter host phenotype by inducing the production of the bacteriocin sublancin, which contributes to antagonism against ancestral prophage-free variants (21). Beyond this antagonistic effect, SPβ also modifies its host through site-specific integration into the *spsM* gene (formerly ypqP), a locus reported to be involved in sporulation and biofilm development (20, 22, 23). The *spsM* gene encodes a polysaccharide synthase involved in the synthesis of legionaminic acid (Leg), a key component of the spore crust that enhances dispersibility by increasing spore surface hydrophilicity (22).

Integration of SPβ into *spsM* results in temporary gene disruption and is reversed through site-specific excision prior to sporulation (20). This reversible integration-excision mechanism, known as active lysogeny, has also been described in other bacterial species, such as *Clostridioides difficile*, *Listeria monocytogenes*, and *Streptococcus pneumoniae*, as well as at additional chromosomal loci in *B. subtilis* (24). Active lysogeny represents a sophisticated prophage-host relationship, where the prophage excises under specific physiological conditions to restore host gene function. During sporulation, SPβ is known to excise precisely from the *spsM* locus in the mother cell, restoring gene integrity and enabling the expression of *spsM*-dependent functions during late developmental stages (20). This process mirrors similar prophage-like excisions in other loci, such as the *skin* element, which is essential for reconstituting the *sigK* gene involved in sporulation (25, 26).

Regulation of host genes through active lysogeny raises important questions about genome stability and the long-term consequences for both the host and the phage itself. Recent studies have suggested that SPβ integration into the *spsM* locus may impair biofilm formation, potentially enhancing phage horizontal transmission within the bacterial population (23, 27). Biofilms are multicellular bacterial aggregates, where bacteria are embedded in an extracellular matrix (ECM), essential for environmental resilience and interspecies interactions (9). They can also block external phage attacks or their transmission within the population (28–30). Given the ecological relevance of biofilms and the complex social traits they entail (1, 31), prophage integration into biofilm-associated loci could represent an adaptive strategy that enables phages to manipulate host development and behavior. Such insertions can modulate biofilm architecture, alter sporulation dynamics, and influence competitive interactions with

closely related strains—ultimately shaping both host fitness and community structure (27).

To better understand how disruptions at the *spsM* locus influence multicellular traits in *B. subtilis*, we examined the phenotypic consequences of distinct modes of disruptions at the *spsM* locus, including SPβ integration, insertional mutagenesis (*spsM::kan*), and a markerless Δ*spsM* deletion. Earlier work suggested that SPβ insertion into *spsM* reduces biofilm formation, raising the question of whether *spsM* inactivation itself has a consistent functional impact across diverse genetic backgrounds. In this study, we therefore assessed the effects of different *spsM* disruptions on biofilm architecture, motility, and spontaneous phenotypic diversification. Although our findings do not support a universal role for *spsM* in biofilm development, they show that specific modes of genetic manipulation at this locus can alter the likelihood of acquiring secondary mutations that strongly influence multicellular behavior.

More broadly, our results emphasize that prophage–host interactions must be interpreted in the context of background-specific regulatory networks and the inherent tendency of certain genes, such as *swrA* and *comP*, to accumulate mutations under selection. The contrasting outcomes associated with SPβ integration and *spsM::kan* disruption highlight how both local genomic context and experimental manipulation can shape the emergence of adaptive variants. These observations underline the importance of routine genomic verification during strain construction and contribute to a more nuanced understanding of how mutation supply and selection interact to drive phenotypic diversification in bacterial populations.

## MATERIALS AND METHODS

### Bacterial strains and growth conditions

Bacterial strains used and constructed in this study are presented in Table 1. Plasmids and oligonucleotides used for strain construction and Sanger sequencing are presented in Tables 2 and 3, respectively. Accession numbers of genomes analyzed with whole-genome sequencing are presented in Table 4. All the strains used were first transferred from −80°C 10% glycerol stock cultures into 5 mL liquid lysogeny broth (LB, Condalab, Spain) media and incubated at 37°C and 220 rpm for 16 h. The cultures were then used for further experiments.

Mutant *B. subtilis* strains prepared in this study were constructed by transforming the desired strains with polymerase chain reaction (PCR) products or with plasmids carrying the desired region (40). Successfully transformed mutants were obtained by selecting for desired antibiotic resistance. Working concentrations of antibiotics were 5 µg/mL kanamycin (Kan), 100 µg/mL spectinomycin (Spec), 5 µg/mL chloramphenicol (Cm), 10 µg/mL tetracycline (Tet), and 100 µg/mL ampicillin (Amp).

The NDmed *spsM::kan* (MP) and PS-216 *spsM::kan* strains were generated by transforming the NDmed wt and PS-216 wt strains with a PCR product of the *spsM::kan* region from the NDmed *spsM::kan* (GM3248) strain. The pIC679 plasmid (Table 2), carrying the *spsM* gene with its native promoter, was previously constructed and described in Sanchez-Vizuete et al. (23). The pIC679 plasmid was transformed into *E. coli* DH5α strain (23, 40), then isolated and used to generate strains prepared in this study, labeled as *amyE::spsM*. Strains labeled as *swrA::tet* or *comP::ery* were constructed by transforming the desired strains with PCR product of the *swrA::tet* or *comP::ery* region from DS215 or BKE31690 strain, respectively. The pDR111::P$_{hs}$-*comP* plasmid (Table 2) was isolated from the strain AEC1002 and was used to construct the NDmed *spsM::kan* (GM3248) *amyE::comP* strain.

Fluorescently labeled strains were constructed by transforming the desired strains with the phy-sGFP or pTB498 plasmids (Table 2), which carry the sGFP and mKate fluorescent marker fused with the hyper-spank promoter, respectively. The NDmed wt, *spsM::kan* (GM3248), and *spsM::kan amyE::spsM* (GM3326) strains were previously constructed and described in Sanchez-Vizuete et al. (23). The P9_B1 fluffy strain was

**TABLE 1** *B. subtilis* and *Escherichia coli* strains used in experiments or as source of gDNA and plasmids for strain construction

| Strain label | Background and genotype | Source or reference |
|---|---|---|
| *B. subtilis* | | |
| DS215 | 3610 *swrA::tet* | (32) |
| BKE17040 | 168 *mutS::ery* | (33) |
| BKE31690 | 168 *comP::ery* | (33) |
| MB8_B7 | MB8_B7 *spsM*::SPβ (natural lysogen) | (34) |
| PS-216 wt | PS-216 wild type | (35) |
| PS-216 *spsM::kan* | PS-216 *spsM::kan* | This work |
| *B. subtilis* P9_B1 | | |
| wt | wild type | (34) |
| DTUB222 | *amyE::mkate* (*spec*) | (21) |
| DTUB43 | *amyE::gfp* (*cm*) | (21) |
| Δ*spsM* | Δ*spsM* (markerless deletion) | This work |
| *spsM::kan* | *spsM::kan* | (21) |
| *spsM::kan amyE::spsM* | *spsM::kan amyE::spsM* (*spec*) | This work |
| Lys SPβ | *spsM*::SPβ | (21) |
| fluffy | *spsM::kan swrA* | This work |
| *swrA::tet* | *swrA::tet* | This work |
| *spsM::kan swrA::tet* | *spsM::kan swrA::tet* | This work |
| fluffy GFP | *spsM::kan swrA amyE::gfp* (*cm*) | This work |
| fluffy mKate | *spsM::kan swrA amyE::mkate* (*spec*) | This work |
| *swrA::tet* GFP | *swrA::tet amyE::gfp* (*cm*) | This work |
| *swrA::tet* mKate | *swrA::tet amyE::mkate* (*spec*) | This work |
| *spsM::kan swrA::tet* GFP | *spsM::kan swrA::tet amyE::gfp* (*cm*) | This work |
| *spsM::kan swrA::tet* mKate | *spsM::kan swrA::tet amyE::mkate* (*spec*) | This work |
| *spsM::kan amyE*::mKate | *spsM::kan amyE::mkate* (*spec*) | This work |
| *B. subtilis* NDmed | | |
| wt | wild type | (36) |
| *spsM::kan* (GM3248) | *spsM::kan*[a] | (23) |
| *spsM::kan amyE::spsM* (GM3326) | *spsM::kan amyE::spsM*[a] (*spec*) | (23) |
| *spsM::kan* (MP) | *spsM::kan* | This work |
| *spsM::kan amyE::spsM* (MP) | *spsM::kan amyE::spsM* (*spec*) | This work |
| *spsM::kan* (GM3248) *amyE::spsM* (MP) | *spsM::kan*[a] *amyE::spsM* (*spec*) | This work |
| Lys SPβ | *spsM*::SPβ | This work |
| *spsM::kan* (GM3248) *amyE::comP* | *spsM::kan*[a] *amyE*::P$_{hs}$-*comP* (*spec*) | This work |
| *spsM::kan* (GM3248) *comP::ery* | *spsM::kan*[a] *comP::ery* | This work |
| *spsM::kan amyE::spsM* (GM3326) *comP::ery* | *spsM::kan amyE::spsM*[a] *comP::ery* | This work |
| *spsM::kan* (MP) *comP::ery* | *spsM::kan comP::ery* | This work |
| *spsM::kan amyE::spsM* (MP) *comP::ery* | *spsM::kan amyE::spsM comP::ery* | This work |
| *E. coli* | | |
| AEC1002 | pDR111::P$_{hs}$-*comP* (*amp*, *spec*) | (37) |
| DH5α pIC679 | pIC679 (*amp*, *spec*) | This work |
| DH5α pMP101 | pMP101 (*kan*) | This work |
| ECE358 | pJOE8999 (*kan*) | (38) |
| MC1061 | phy-sGFP (*cm*) | (21) |
| MC1061 | pTB498 (*spec*) | (39) |

[a]The genotype listed is as described in the research article by Sanchez-Vizuete et al. (23).

obtained as a result of a spontaneous mutation in P9_B1 *spsM::kan* strain grown as macrocolonies on LB agar plates after 48 h of incubation at 30°C.

The NDmed Lys SPβ strain was constructed by preparing a SPβ phage suspension from the MB8_B7 strain, by prophage induction using 0.5 µg/mL mitomycin C. The phage

**TABLE 2** Plasmids used for *B. subtilis* mutant construction

| Plasmid | Description | Source or reference |
|---|---|---|
| pDR111::P$_{hs}$-*comP* | pDR111 derivative containing *comP* (*amp*, *spec*) with IPTG inducible hyper-spank (hs) promoter | (37) |
| pIC679 | pDR111 derivative containing *spsM* (*amp*, *spec*) with native promoter | (23) |
| pJOE8999 | Plasmid carries Cas9 under the control of mannose-inducible promoter and a sgRNA-encoding sequence transcribed via a strong promoter | (38) |
| phy-sGFP | Plasmid carrying sGFP with hs promoter for fluorescent labeling | (21) |
| pMP101 | pJOE8999 derived plasmid carrying the *spsM* targeting sgRNA and the upstream/downstream homology arms for Δ*spsM* markerless deletion | This work |
| pTB498 | Plasmid pWK-Sp carrying mKate with hs promoter for fluorescent labeling | (39) |

suspension was then used to lysogenize the NDmed wt strain. The presence of SPβ phage in the *spsM* gene was confirmed by PCR using the primer pairs presented in Table 3. The presence of the phage in the genome was additionally confirmed by Illumina whole-genome sequencing (Table 4). NDmed Lys SPβ was additionally verified through prophage activity test by 0.5 µg/mL mitomycin C induction, followed by plaque assay, as previously described (21). Lysogeny was additionally confirmed with sublancin activity assay, as previously described (41).

To prepare the markerless Δ*spsM* deletion mutant of *B. subtilis* P9_B1 strain, the regions upstream and downstream of *spsM* were PCR-amplified from P9_B1 wild-type genomic DNA using the primer pairs oMP19/oMP20 and oMP21/oMP22, respectively. The pJOE8999 backbone (38) was PCR-amplified using the primer pairs oMP15/oMP16-2 and oMP17-2/oMP18. Primers oMP16-2/oMP17-2 introduced the 20-nucleotide *spsM*-targeting guide (spacer) sequence into the pJOE8999 sgRNA expression cassette via primer overhangs. All PCR fragments were purified and assembled using the NEBuilder HiFi DNA Assembly Kit (New England Biolabs) to generate pMP101 (pJOE8999-ΔspsM), a pJOE8999-derived editing plasmid carrying the *spsM* targeting sgRNA and the upstream/downstream homology arms for markerless deletion. The plasmid construction was verified via PCR using the primer pair D-spsM-R/U-spsM-F. The P9_B1 Δ*spsM* markerless mutant was obtained by transforming pMP101 into *B. subtilis* P9_B1 wild-type strain, as described previously (42). The Δ*spsM* deletion was confirmed by PCR using the primer pair D-spsM-R/U-spsM-F and Sanger sequencing (Macrogen, Maastricht).

## Standardized plate preparation

A standardized protocol of solid media preparation was developed to ensure reproducibility of colony morphology across replicates. The medium was always prepared one day prior to inoculation using 250 mL Erlenmeyer flasks containing either 100 mL or 200 mL of medium with the desired amount of agar. Following autoclaving, the medium was cooled to 55°C for 30 min (100 mL) or 45 min (200 mL), and 20 mL aliquots were then transferred into 90 mm Petri dishes. To ensure similar humidity for all plates, they were left at room temperature to dry overnight without stacking. Immediately before inoculation, plates were dried in a laminar flow hood for a specific duration tailored to each experimental protocol.

## Macrocolony experimental conditions and image analysis

To compare macrocolony morphologies among *B. subtilis* mutant strains, cultures were grown on various solid media containing 1.5% agar that were prepared using the standardized plate preparation protocol described above. Colony morphology was

**TABLE 3** Oligonucleotides used in this work[a]

| Label | Fw/Rev | Sequence (5′ to 3′) | Purpose |
|---|---|---|---|
| amyEF | Fw | TCTCCAGTCTTCACATCGGTTTG | Verify insertion into *amyE* locus |
| amyER | Rev | GCAAGAGAAAAGTTTTGTCTGATTTATG | Verify insertion into *amyE* locus |
| comPF | Fw | GCGTTAATCTCCAAACCAACA | Amplify *comP::ery* for strain construction |
| comPF2 | Fw | GCTCCTCAATATGCAGGACATT | Detect *comP* mutations (Sanger seq.) |
| comPR | Rev | ACAAGATGCAAGACCTAATTAACTAC | Amplify *comP::ery* for strain construction |
| comPR2 | Rev | GCATTAGCTTCGGCGTCAT | Detect *comP* mutations (Sanger seq.) |
| oAD1 | Fw | ATCTGGACTGGCACCTTATGGATACC | Check for SPβ integration with oAD3 |
| oAD2 | Rev | CTGCTCTGGAAAGGAAGGCAGAGTAA | Check for SPβ integration with oTB122 |
| oAD3 | Rev | ATGACCGAACCTCTGGAACCGAGAAC | Check for SPβ integration with oAD1 |
| oAD61 | Fw | GCAACATGTGCCTGCTGAAG | Confirm presence of sublancin cluster |
| oAD62 | Rev | GGTATGCCATATGCTCAACC | Confirm presence of sublancin cluster |
| oMP3 | Fw | ATGATGACCTTATGCCTTTTTCTCTTGATGCAATTC | Amplify *spsM* locus (confirm intact *spsM*, presence of *spsM::kan* etc.) |
| oMP4 | Rev | CATGCAAGCATGCCTAAACAGCAAACAGC | Amplify *spsM* locus (confirm intact *spsM*, presence of *spsM::kan* etc.) |
| oMP5 | Fw | CCCGTTTGATTCTTACATAGCCGAA | Amplify *swrA::tet* for strain construction, Detect *swrA* mutations (Sanger seq.) |
| oMP6 | Rev | GCGAGAATGCTTGTTAGCACTCC | Amplify *swrA::tet* for strain construction, Detect *swrA* mutations (Sanger seq.) |
| oMP15 | Fw | GTTAATACGTGGGGCCAATAAGGCCTTTCTAG | Amplify pJOE8999 |
| oMP16-2 | Rev | TAGCTCTAAAACAGTGCAGCATGTCATTAATACGTAGGTACATTTTACTCAATTCTC | Amplify pJOE8999, contains *spsM*-targeting guide (spacer) sequence |
| oMP17-2 | Fw | AAATGTACCTACGTATTAATGACATGCTGCACTGTTTTAGAGCTAGAAATAGCAAGTTAAAATAAG | Amplify pJOE8999, contains *spsM*-targeting guide (spacer) sequence |
| oMP18 | Rev | ATAGCAATGAACGCGTTGGCCGTCGACCCTATAG | Amplify pJOE8999 |
| oMP19 | Fw | CGACGGCCAACGCGTTCATTGCTATGAACGATTTTTTTATTC | Amplify region upstream of *spsM* |
| oMP20 | Rev | TGCAATTCTTCACTGTTTAGGCATGGCTTATC | Amplify region upstream of *spsM* |
| oMP21 | Fw | CATGCCTAAACAGTGAAGAATTGCATCAAGAGAAAAG | Amplify region downstream of *spsM* |
| oMP22 | Rev | GGCCTTATTGGCCCCACGTATTAACAAAGGAGG | Amplify region downstream of *spsM* |
| oTB122 | Fw | TATTGAGCTTGCCAAACTCATAAGAATGAA | Check for SPβ integration with oAD2 |
| D-spsM-R | Rev | GGCCTTATTGGCCGCCGTTCAATGGAAAATCAT | Confirm Δ*spsM* deletion |
| U-spsM-F | Fw | GGCCAACGAGGCCTAAACAAAAGGAGCTTGCGG | Confirm Δ*spsM* deletion |

[a]Fw, forward primers; Rev, reverse primers.

assessed on LB agar (Condalab, Spain), TSA (Condalab, Spain), and LBGM agar, which enhances extracellular matrix production in *B. subtilis* biofilm (43–45). LBGM medium was prepared by supplementing LB agar with sterile 1% glycerol and 100 µM MnSO$_4$ post-autoclaving. For strains harboring constructs under the control of hyper-spank promoter, the media was supplemented with 0.1 mM IPTG. To initiate macrocolony growth, 10 µL of overnight culture was spotted onto the plates, dried for 5 min post-inoculation, and incubated at 30°C for 48 h. To ensure consistency in macrocolony morphology across conditions, different strains were grown on the same plates, and each condition was tested at least in duplicate.

Macrocolony images were acquired using Leica MZ FLIII and Nikon SMZ25 stereomicroscopes at the infrastructural center "Microscopy of Biological Samples" (Biotechnical Faculty, University of Ljubljana, Slovenia). Surface structure of macrocolony biofilms was analyzed with SurfCharJ plugin (46) in Fiji (ImageJ) (47). Pixel intensity in the 2D images was assumed to correlate with object height, with brighter regions representing more protruding surface features. The colony morphology was analyzed by pairwise correlation analysis implemented in a custom C++ program routine (48). First, by using Fiji (ImageJ), the RGB image was split into three color channels. For further analysis, we have always chosen the green channel image, where the background of the colony was subtracted. Then, all non-background pixel intensity values were exported, together

**TABLE 4** Strains analyzed with whole-genome sequencing and their short read archive accession numbers

| Strain | Accession numbers | Sequencing method |
| --- | --- | --- |
| P9_B1 *spsM::kan* | SRR31313896 | Illumina |
| P9_B1 fluffy | SRR31346685 | Illumina |
| NDmed wt | Raw reads: SRR32479380 | PacBio |
| | Assembled (annotated): CP183238 | |
| NDmed *spsM::kan* (GM3248) | SRR32463897 | Illumina |
| NDmed *spsM::kan amyE::spsM* (GM3326) | SRR31379056 | Illumina |
| NDmed *spsM::kan* (MP) | SRR32465634 | Illumina |
| NDmed Lys SPβ | SRR32478187 | Illumina |

with corresponding positional coordinates. These served as input data to a C++ program routine that computes the normalized pairwise correlation function, which is defined as:

$$\gamma(r) = \langle I(x)I(y)\rangle / \langle I^2 \rangle \tag{1}$$

where $< I(x)I(y)>$ represents the average product of the intensities of two pixels separated by a distance $r$, and $<I^2>$ refers to the average of the squared intensities across all pixels. This normalization results in $\gamma(0)$ equal to 1. In images containing larger objects, the pairwise correlation function decays more slowly with distance. Oscillations in the function reflect repeating spatial motifs and local ordering. In contrast, images composed of randomized, unstructured pixels exhibit a rapid decay of the correlation function toward zero, with no undulations.

The macrocolony morphology of original stocks of the NDmed *spsM::kan* mutants (23), GM3248 clone 1 and GM3248 clone 2, was tested by the research team at the Micalis Institute. To perform the macrocolony assay, 6-well plates were prepared by adding 4 mL of LB (BD Difco, USA) with 1.5% agar or TSA (BioMérieux, France) with 1.5% agar to each well. The plates were dried under a laminar flow hood for 1.5 h and then left at room temperature for 5 days prior to use. For inoculum preparation, a single colony was grown overnight in 5 mL of LB or TSB, supplemented with kanamycin (8 µg/mL) for the mutant strains, at 30°C with shaking at 180 rpm for 18 h. Cultures were centrifuged at 5,000 × *g* for 5 min and resuspended in fresh medium. A 4 µL aliquot of the resuspension was spotted at the center of each well, and plates were incubated at 30°C to allow macrocolony development.

## Swimming and swarming motility assays

Swimming and swarming motility of *B. subtilis* strains were assessed on plates prepared using the standardized plate preparation protocol described above. Swimming motility was tested on LB soft agar plates containing 0.3% agar, while swarming motility was assessed on LB semi-solid agar plates with 0.7% agar, using an adjusted version of a previously described protocol (49). For strains harboring constructs under the hyper-spank promoter, the media was supplemented with 0.1 mM IPTG. Plates were prepared one day prior to use and dried overnight at room temperature without stacking. Immediately before inoculation, 0.3% agar plates were dried in a laminar flow hood for 5 min, and 0.7% agar plates were dried for 20 min to standardize surface moisture. Two microliters of culture were spotted at the center of the prepared plates, which were then incubated at 37°C for up to 8 h without stacking, to maintain uniform humidity. Motility was quantified as the mean colony diameter measured along two perpendicular axes.

## Competition assay

The competition assay was conducted using standardized macrocolony preparation as described above. Strain mixtures were grown on LB and LBGM solid media (1.5% agar) prepared using the standardized plate preparation protocol described above. Overnight cultures of *B. subtilis* P9_B1 strains were adjusted to equal optical density at 600 nm ($OD_{600}$). Strain pairs, each tagged with either GFP or mKate fluorescent reporters, were mixed in equal volumes (100 µL each; 1:1 [vol/vol]). From each mixture, 10 µL was spotted onto LB or LBGM agar plates and incubated at 30°C for 48 h. To correct for potential differences in reporter fluorescence intensity, monocultures of each fluorescent strain were grown in parallel on the same medium.

Macrocolony images were acquired using a Nikon SMZ25 stereomicroscope at the infrastructural center "Microscopy of Biological Samples" (Biotechnical Faculty, University of Ljubljana, Slovenia). Fluorescence quantification was performed in Fiji (ImageJ) (47) by measuring the integrated density of the green (GFP) and red (mKate) channels. Values from cocultures were normalized against monoculture fluorescence intensities to account for potential variability in reporter expression. Relative abundance of each strain in coculture was expressed as a percentage of the total normalized fluorescence signal.

## Spontaneous mutation screening

To assess whether disruption of the *spsM* gene promotes spontaneous diversification and the emergence of mutations, we conducted a screening assay based on conditions under which such mutations were first observed. A volume of 10 µL from overnight cultures was spotted onto LB agar plates (Condalab, Spain), briefly dried for 5 min, and incubated at 30°C for 48 h. All strains were plated on the same LB agar plate as separate macrocolonies to minimize variability in growth conditions.

Following incubation, peripheral outgrowths that developed at the edges of macrocolonies and displayed morphology distinct from that of colony center were identified. Both peripheral and central regions were restreaked on fresh LB agar plates, and colony morphology was assessed after incubation. Outgrowths that consistently showed different morphology from the central colony were grouped by visual phenotype. A subset of these strains, representing each morphological group, was selected for Sanger sequencing of the *swrA* and *comP* loci using the primer pairs oMP5/oMP6 and comPF2/comPR2 (Table 3). Within each group, strains exhibited identical sequencing results, allowing classification based on shared morphological features. One representative strain from each group was chosen for further phenotypic analysis, including assessment of macrocolony morphology and swimming and swarming motility, as described above.

The assay was repeated across four independent experiments. Minor differences in environmental conditions, such as humidity and LB agar batch, may have influenced the frequency of variant emergence. Therefore, results from each experiment are presented independently. We did not perform statistical comparisons or aggregate the data across replicates, as the purpose of this analysis was to qualitatively document the consistent association between *spsM* disruption and spontaneous mutation emergence.

To determine whether the frequency of visible outgrowths could be explained by differences in population size among strains, CFU/mL counts were obtained from macrocolonies grown under the same conditions used for the spontaneous mutation screening. After 48 h of incubation at 30°C, the entire biomass of each macrocolony was scraped and resuspended into 1,000 µL of 0.9% NaCl, then homogenized by vortexing for 5 min, followed by sonication on ice for 5 min (5 s ON, 5 s OFF, 20% amplitude) to disperse cell aggregates. Complete dispersion was verified microscopically. Serial tenfold dilutions were plated on LB agar and incubated overnight at 37°C. Colony counts were used to calculate CFU/mL for each strain.

## Fluctuation assay

Mutation rates were estimated using a rifampicin-based fluctuation assay adapted from a previously established protocol (50, 51). Single colonies of each strain were obtained by streaking from −80°C glycerol stocks onto LB agar and incubating overnight at 37°C. Individual colonies were used to inoculate a 96-well microtiter plate containing 200 μL LB medium per well. Plates were incubated in a multimode microplate reader (BioTek Synergy H1) at 37°C with linear continuous shaking (3 mm), and $OD_{600}$ was measured every 5 min for 24 h.

To determine population size, independent wells of each strain were selected for CFU determination. Cultures were serially diluted in 0.9% NaCl and plated onto LB agar. Plates were incubated overnight at 37°C, and colonies were counted to calculate CFU/mL. The remaining wells were used to quantify rifampicin-resistant mutants. The entire 200 μL culture volumes were spread onto LB agar plates supplemented with rifampicin (100 μg/mL). Plates were incubated overnight at 37°C, and the number of rifampicin-resistant colonies was counted. Because rifampicin resistance in *B. subtilis* arises primarily through mutations in the *rpoB* gene, the number of resistant colonies reflects the occurrence of spontaneous mutations during outgrowth. Mutation rates were estimated in RStudio using the *flan* package, which applies maximum likelihood inference under the Lea–Coulson model of the Luria–Delbrück distribution. For each strain, the distribution of rifampicin-resistant colonies, together with the mean final population size, was used to estimate the mutation parameter and convert it to a per-cell mutation rate. Differences between strains were assessed using *flan's* built-in two-sample likelihood ratio test.

## Whole-genome sequencing and Sanger sequencing

Whole-genome sequencing (WGS) of *B. subtilis* P9_B1 and NDmed mutant strains was performed by SeqOmics Biotechnology Ltd. using an Illumina NextSeq platform. Paired-end libraries (2 × 150 bp, ~10 million reads per sample, ~700× coverage) were prepared using the NEBNext Ultra II DNA Library Prep Kit and sequenced with the NextSeq 500/550 High Output Kit v2. Reads were mapped to the corresponding reference genome assemblies (NDmed: ASM74047v1 and ASM4853740v1; P9_B1: ASM966245v1), and single-nucleotide polymorphism (SNP) analysis was conducted using CLC Genomics Workbench 23.0.4.

For NDmed strains, initial mapping used the draft reference genome assembly (ASM74047v1, 10 contigs). To improve resolution, a complete circular genome of NDmed wild type was generated *de novo* using PacBio Sequel II (Novogene Europe) and annotated with Prokka (52). High-quality Illumina reads were aligned to this reference using Bowtie (53), and SNPs were identified using SAMtools (54), following quality checks with MultiQC (55). Detected SNPs in *swrA* and *comP* were validated via PCR and Sanger sequencing (Macrogen, Maastricht) using the primer pairs oMP5/oMP6 and comPF2/comPR2 (Table 3).

## Bioinformatic and statistical analysis

Statistical analysis for all measurements was performed using ANOVA and Tukey's Honest Significant Difference method ($P < 0.05$) in RStudio with the aov() and TukeyHSD() functions, respectively. The multcompLetters4() function was used to obtain the compact letter display from the performed ANOVA. The ExPASy translate tool (56) was used to translate the DNA sequences acquired with Sanger sequencing to analyze their potential effect on translated protein sequences. All nucleotide sequences were viewed, analyzed, and aligned using Snapgene (GSL Biotech) with the algorithm MUSCLE (MUltiple Sequence Comparison by Log-Expectation) (57, 58).

To show genetic similarity between *B. subtilis* P9_B1 and *B. subtilis* NDmed strains, a phylogenetic tree was obtained using the autoMLST2 (59) free web server for microbial phylogeny and taxonomy, in Placement (Fast) mode. Next to P9_B1 and NDmed, 11 genomes of other *B. subtilis* strains (each containing SPβ prophage) were included in

the analysis, while the remaining strains present on the tree were automatically assigned by the software. The tree was exported as an nwk file and visualized using iTOL (60). The average nucleotide distance between P9_B1 and NDmed was calculated using the FastAni algorithm (61).

## RESULTS

### Emergence of new morphotype observed in *B. subtilis* P9_B1 with disruption of prophage integration locus *spsM*

The primary objective of this study was to investigate how SPβ integration and other disruptions of the *spsM* locus affect biofilm formation in *B. subtilis* and to explore whether these alterations produce broader ecological or evolutionary effects under laboratory conditions. Previous research indicated that insertion of the SPβ prophage, which is present in common laboratory strains such as *B. subtilis* 168 and NCBI 3610, disrupts *spsM* and negatively affects biofilm formation. Consistent with this idea, the clinical isolate NDmed, which lacks SPβ and carries an intact *spsM* gene, was shown to form robust biofilms (23).

To evaluate the impact of *spsM* disruption on colony morphology, macrocolonies of different *B. subtilis* strains were repeatedly grown on standardized solid LB medium. Sanchez-Vizuete et al. (23) reported that *spsM::kan* mutation in NDmed strain leads to a flat phenotype characterized by smooth, unstructured colonies that lack the typical wrinkled architecture. We first re-examined this observation and were able to reproduce the reported flat morphology using the NDmed *spsM::kan* (GM3248) strain (Fig. 1A). We then tested whether the same phenotype would appear in the soil isolate P9_B1, which also carries an intact *spsM* gene and lacks the SPβ prophage. The *spsM::kan* mutation introduced into P9_B1 was identical to the mutation present in NDmed *spsM::kan* (GM3248) strain. Despite this, the mutation did not produce the smooth, unstructured morphology observed in NDmed. Instead, colonies retained the typical wrinkled architecture (Fig. 1A). The contrasting effects of the same *spsM::kan* mutation in NDmed and P9_B1 are examined in more detail in the following sections. To further explore the role of *spsM* disruption, we also lysogenized both NDmed and P9_B1 with the SPβ prophage. Neither lysogen formed the flat colony morphology described for the NDmed *spsM::kan* (GM3248) strain (Fig. 1A). Confirmation of lysogenization for NDmed is presented in Fig. S1, while the lysogenic state of P9_B1 has been previously verified (21).

Interestingly, on solid LB medium, instead of displaying the expected morphology changes, the P9_B1 *spsM::kan* strain occasionally developed peripheral outgrowths that were absent in the wild-type strain (Fig. 1B), suggesting spontaneous diversification through the appearance of mutations. These outgrowths typically appeared after 48 h of incubation at 30°C. To determine whether the outgrowths resulted from a spontaneous mutation, we transferred both the center and the outgrowth of a single macrocolony onto fresh solid LB medium and examined the resulting colony morphologies (Fig. 1B). The macrocolonies derived from the outgrowth displayed a distinct morphology compared to both the original P9_B1 *spsM::kan* strain and the central region of the same macrocolony. This newly emerged lineage of *spsM::kan* was named "fluffy," due to an irregular colony edge resembling a cotton wool. The "fluffy" morphotype was stable and could be recreated from −80°C stock, followed by overnight culture and colony spotting, indicating it was caused by an additional mutation that emerged in the *spsM::kan* strain background (Fig. 1B).

### "Fluffy" morphotype is caused by spontaneous mutation in *swrA* which rapidly arises in *spsM::kan* genetic background

Illumina whole-genome resequencing was performed on genomic DNA of both the ancestral P9_B1 *spsM::kan* strain (NCBI SRA accession number: SRR31313896) and the morphologically distinct spontaneously derived mutant strain, hereafter referred to as P9_B1 fluffy (NCBI SRA accession number: SRR31346685). A single mutation unique to

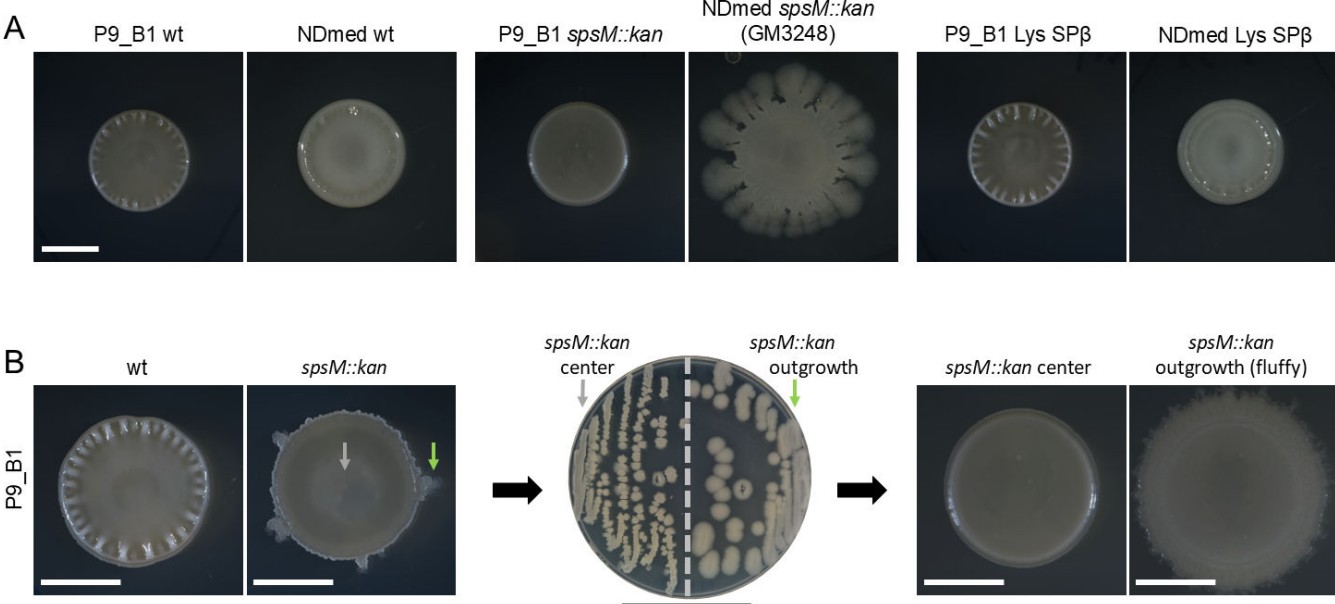

**FIG 1** Morphological comparison of *B. subtilis* macrocolonies grown on standardized solid LB medium. (A) Macrocolony morphology of P9_B1 and NDmed strains grown on LB agar at 30°C for 48 h. Scale bar: 5 mm. (B) Macrocolony morphology of the P9_B1 *spsM::kan* strain and its derivatives. All strain labels refer to P9_B1 strains. On the left, macrocolonies of the wild type (wt) and *spsM::kan* mutant are shown after 48 h of growth on LB agar at 30°C. The middle panel shows an LB plate with isolated colonies derived from streaking the center (gray arrow) and outgrowth (green arrow) regions of the *spsM::kan* macrocolony. On the right are macrocolonies formed by the isolated strains: the *spsM::kan* center and the outgrowth strain referred to as "fluffy." White scale bar: 5 mm. Black scale bar: 5 cm.

the P9_B1 fluffy strain was identified: a thymine deletion at position 26 of the *swrA* coding sequence (*swrA*:c.26delT), which shortens a homopolymeric T tract from eight to seven nucleotides and results in a frameshift mutation (Fig. 2). Mutations within this tract are known targets of slipped-strand mispairing, and similar frameshift alterations have been reported as domestication mutations in laboratory strains such as *B. subtilis* 168, which carries a nine-T variant that also disrupts *swrA* (32, 62, 63). These findings indicate that the P9_B1 *spsM::kan* strain went through a fast spontaneous genetic diversification within 48 h of growth on solid LB medium at 30°C.

To assess whether the spontaneous *swrA* mutation in the P9_B1 fluffy strain affects macrocolony biofilm morphology, we examined macrocolonies on solid LBGM medium (Fig. 2), which promotes biofilm formation by enhancing extracellular matrix production (43–45). The morphology of P9_B1 fluffy was considerably different from that of its ancestral P9_B1 *spsM::kan* strain. Macrocolony biofilm of P9_B1 fluffy was more widely spread, flatter, and exhibited smaller wrinkles (Fig. 2). To confirm these changes in morphology resulted from the *swrA* frameshift mutation, we constructed *swrA* disruption mutants P9_B1 *swrA::tet* and P9_B1 *spsM::kan swrA::tet*. P9_B1 fluffy and P9_B1 *spsM::kan swrA::tet* mutants exhibited indistinguishable morphologies, supporting the conclusion that the *swrA* mutation leads to altered macrocolony biofilm morphology in P9_B1 fluffy.

## Spontaneous mutation in *swrA* in the P9_B1 fluffy strain leads to loss of swarming and impaired swimming motility

In *B. subtilis*, SwrA regulates both swimming and swarming by promoting flagellar biosynthesis (64). Since the P9_B1 fluffy strain resulted from a spontaneous mutation in *swrA*, we compared motility of wild-type and mutant strains on LB medium containing 0.3% (swimming) and 0.7% (swarming) agar (Fig. 3). P9_B1 wt and *spsM::kan* strains displayed similar swimming and swarming motility patterns. During swimming, both strains formed uniform, featureless colonies, whereas under swarming conditions, they

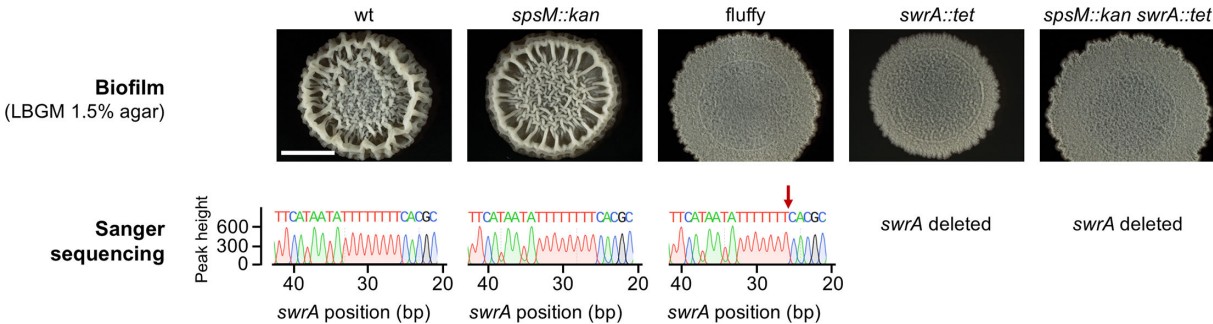

**FIG 2** Biofilm morphology and Sanger sequencing confirmation of a spontaneous *swrA* mutation in the P9_B1 fluffy strain. All strain labels refer to P9_B1 derivatives. The "fluffy" strain refers to the *spsM::kan* strain carrying the spontaneous *swrA* c.26delT mutation. The top panel shows macrocolony biofilms of strains with (wt and *spsM::kan*) and without (fluffy, *swrA::tet*, *spsM::kan swrA::tet*) an active *swrA*. Macrocolony biofilms were grown on solid LBGM medium and imaged after 48 h of incubation at 30°C. Scale bar: 5 mm. The bottom panel displays segments of Sanger sequencing results covering positions 20–40 bp of the *swrA* gene. The nucleotide sequences derived from Sanger reads are shown for each strain, with the corresponding peak height plotted on the Y-axis. The spontaneous swrA:c.26delT mutation in the fluffy strain is indicated by a red arrow and corresponds to a frameshift mutation not observed in the other strains.

developed structured patterns with compact zones and dendritic extensions. In contrast, all three strains carrying the *swrA* mutation were significantly impaired in both modes of motility. Swimming motility was still present in *swrA* mutants but was statistically significantly lower than in strains with intact *swrA* (Fig. 3B and D). Notably, the swimming patterns of P9_B1 fluffy and *spsM::kan swrA::tet* strains were inconsistent across replicates, sometimes displaying thin dendrites and other times not. In contrast, all *swrA* mutant strains completely lost the ability to swarm (Fig. 3B and C). These results confirm that the spontaneous *swrA* frameshift mutation in P9_B1 fluffy leads to a loss of swarming motility and altered macrocolony morphology.

## Flat morphotype in *B. subtilis* NDmed *spsM::kan* (GM3248) is associated with spontaneous mutation in *comP*

We further investigated why disruption of *spsM* (*spsM::kan*) leads to pronounced changes in biofilm morphology in the *B. subtilis* clinical isolate NDmed but not in the soil isolate P9_B1, and why we were unable to recreate the flat colony morphotype in the SPβ lysogen of NDmed. In order to do that, we tested the effects of *spsM* complementation, recreated *spsM* deletion in the NDmed strain, and reexamined macrocolony biofilms on other media types, such as TSA and LBGM (Fig. 4). NDmed strains labeled as wt, *spsM::kan* (GM3248), and *spsM::kan amyE::spsM* (GM3326) were previously described by Sanchez-Vizuete et al. (23), while strains labeled with (MP) were constructed as part of this study to independently verify the phenotypic effects of the *spsM::kan* mutation in the NDmed background.

As expected, based on our initial results on LB, we were again able to successfully reproduce previous findings on TSA and LBGM, where NDmed *spsM::kan* (GM3248) strain formed wider, flatter macrocolonies compared to the NDmed wt strain, which formed smaller macrocolonies with prominent large wrinkles (Fig. 4B). In addition, the NDmed *spsM::kan amyE::spsM* (GM3326) strain showed successful complementation, displaying a macrocolony morphology identical to that of the NDmed wt strain, consistent with the results reported by Sanchez-Vizuete et al. (23). In contrast, the NDmed mutant strains constructed in this study (labeled MP) behaved differently. The NDmed *spsM::kan* (MP) strain exhibited no major morphological differences from the NDmed wt strain, nor did the complemented NDmed *spsM::kan amyE::spsM* (MP). We also constructed a *spsM::kan* mutant of *B. subtilis* soil isolate PS-216 and observed no effect on biofilm morphology (Fig. S2A). Additionally, pellicle biofilms were examined across all strains, but no major differences in biofilm morphology were detected (Fig. S2B).

To quantitatively assess whether the *spsM::kan* mutation affects biofilm morphology, we characterized the surface structure and spatial segregation of P9_B1 and NDmed

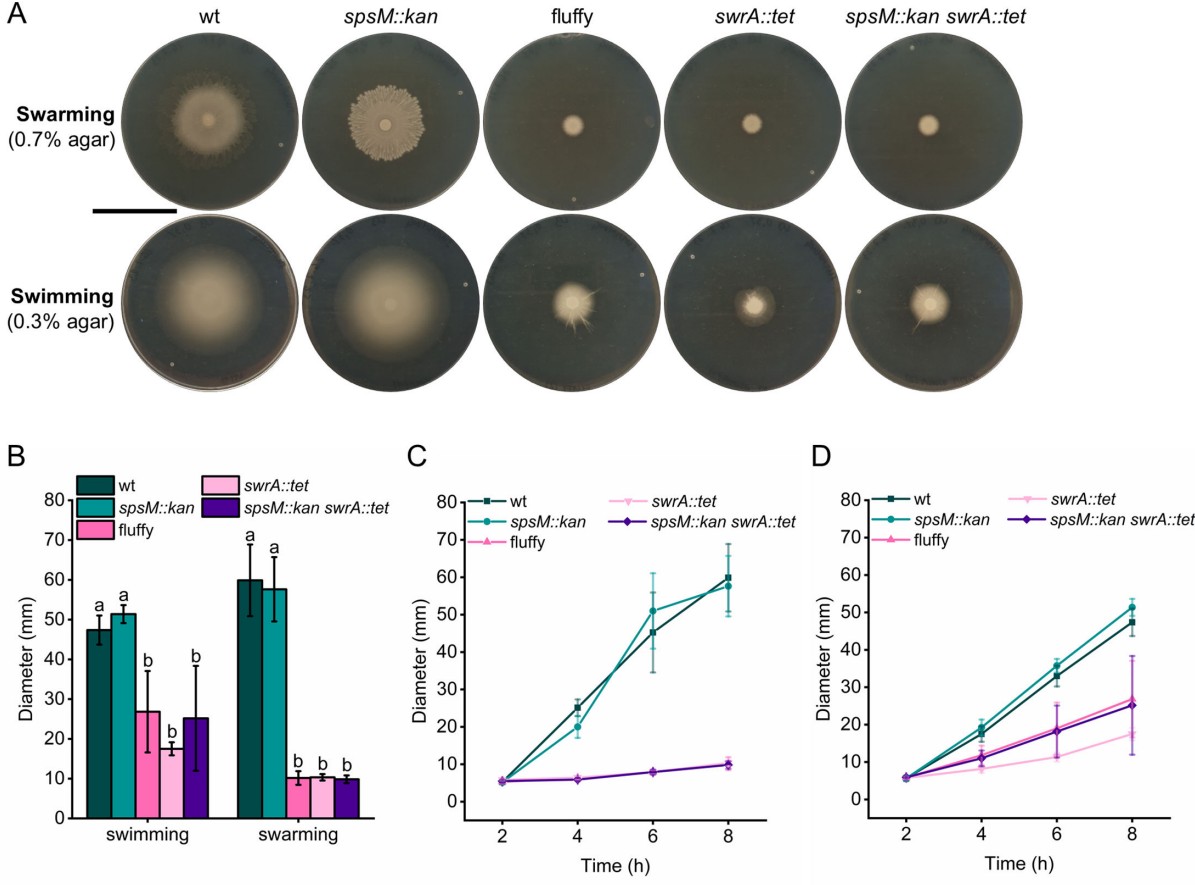

**FIG 3** Decreased swimming and loss of swarming motility resulting from a spontaneous mutation of the *swrA* gene in the P9_B1 fluffy strain. All strain labels refer to P9_B1 derivatives. The fluffy strain refers to the *spsM::kan* strain carrying the spontaneous *swrA* c.26delT mutation. (A) Images of swarming and swimming motility of strains with (wt and *spsM::kan*) and without (fluffy, *swrA::tet*, *spsM::kan swrA::tet*) an active *swrA*. Swimming assays were performed on LB with 0.3% agar and swarming assays on LB with 0.7% agar. Swimming and swarming motility plates were imaged after 8 h of incubation at 37°C. Plates were imaged against a black background so that bacterial colonization zones appear white, and uncolonized agar appears dark. Scale bar: 5 cm. (B) Quantification of swimming and swarming diameters after 8 h at 37°C. Columns represent mean diameters (*n* = 6), and error bars represent standard deviation. Statistical analysis was performed using one-way ANOVA followed by Tukey's Honest Significant Difference (HSD) test. Different letters above the error bars indicate statistically significant differences between strains separately for swimming and swarming (*P* < 0.05). (C and D) Dynamics of swarming (C) and swimming (D) motility over an 8-hour period, with measurements taken at 2 h, 4 h, 6 h, and 8 h.

macrocolony biofilms grown on LBGM medium. Among all the strains tested, only the NDmed *spsM::kan* (GM3248) mutant showed statistically significant differences in surface features, specifically in maximum profile height (Rt), surface area in relative units (SA), and profile skewness (Rsk) (Fig. 4C through F). Compared to the other NDmed and P9_B1 strains, which formed smaller, highly wrinkled macrocolonies, the NDmed *spsM::kan* (GM3248) mutant produced wider and flatter macrocolonies with a lower maximum height (Rt = 1.095 mm vs ~1.3 mm), increased surface area (SA = 2.2 vs ~1.3), and markedly reduced skewness (Rsk = −0.015 vs 2.0–2.7). These measurements suggest a shift from prominent large-scale wrinkling to a higher density distribution of fine-scale surface structures. In contrast, the NDmed and P9_B1 strain *spsM::kan* mutants constructed for this study showed only minor effects on Rsk, none of which were statistically significant compared to their respective wild-type strains. A similar trend was observed in pairwise correlation analyses. Only the NDmed *spsM::kan* (GM3248) strain displayed a noticeably different pairwise correlation function compared to all other spsM::kan mutants (Fig. 4G and H). Its correlation function is almost devoid of any oscillations that are present in other strains, suggesting significantly different structural order. Together,

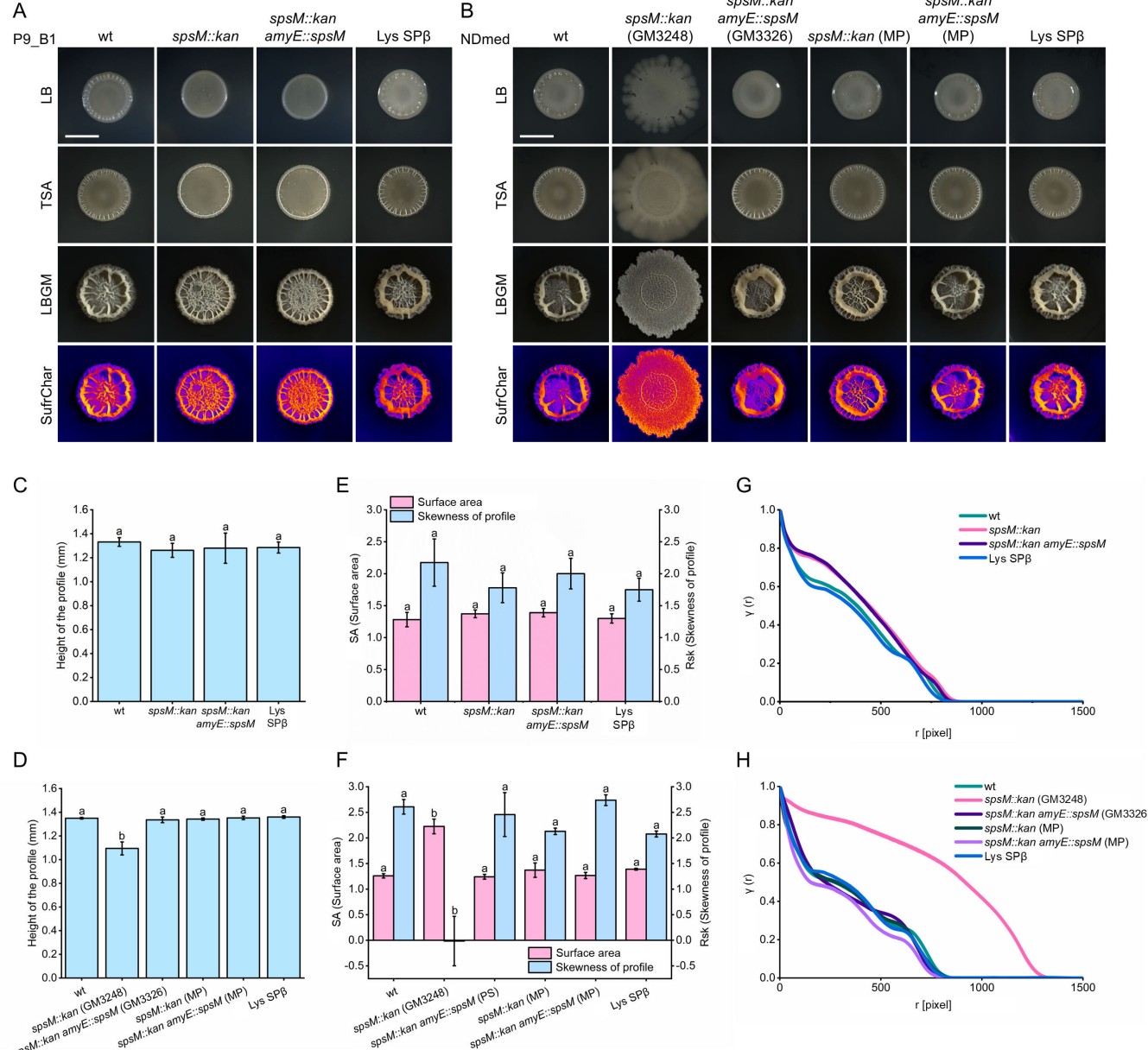

**FIG 4** Macrocolony biofilm morphology, surface characterization, and pairwise correlation analysis of *B. subtilis* P9_B1 and NDmed strains. The NDmed strains labeled as wt, *spsM::kan* (GM3248), and *spsM::kan amyE::spsM* (GM3326) were kindly provided by Sanchez-Vizuete et al. (23). Strains marked with (MP) were constructed in this study to independently confirm the phenotypic effects of the *spsM::kan* mutation in the NDmed background. Images shown are representative of multiple independent experiments. (A and B) Macrocolony biofilm morphology of P9_B1 strains (A) and NDmed strains (B) grown on LB, TSA, and LBGM media at 30°C for 48 h. For LBGM-grown biofilms, surface characteristics were further analyzed using the SurfCharJ plugin in Fiji (ImageJ). Scale bar: 5 mm. (C–F) Quantitative analysis of biofilm structure for P9_B1 (C and E) and NDmed (D and F) strains based on SurfCharJ analysis: (C and D) height of the biofilm profile; (E and F) surface area in relative units and skewness of the biofilm profile. Columns represent mean values (P9_B1: *n* = 6; NDmed: *n* = 4), and error bars indicate standard deviation. Statistical analysis was performed using one-way ANOVA followed by Tukey's Honest Significant Difference (HSD) test. Different letters above error bars denote statistically significant differences between strains (*P* < 0.05). (G and H) Pairwise correlation analysis of macrocolony biofilms grown on LBGM media, performed for P9_B1 (G) and NDmed (H) strains.

the surface characterization and pairwise correlation analysis in Fig. 4 suggest that disruption of *spsM*, whether by antibiotic cassette or prophage integration, does not alter the biofilm phenotype of *B. subtilis*. These findings differ from earlier reports of a significant role for *spsM* in biofilm morphology in the NDmed strain (23).

A further interesting observation was that the biofilm morphology of NDmed *spsM::kan* (GM3248) strain (Fig. 4) was similar to that of the spontaneously mutated P9_B1 fluffy strain (Fig. 2). Both strains show flatter, more widely spread macrocolony biofilm with smaller, more densely distributed wrinkles. This similarity led us to suspect that the NDmed *spsM::kan* (GM3248) strain might carry an additional spontaneous mutation contributing to the altered morphology. To investigate this, the NDmed strains were sent for whole-genome sequencing (NCBI SRA accession numbers: SRR31379056, SRR32463897, SRR32465634, SRR32478187, and SRR32479380). *De novo* sequencing and genome assembly of the NDmed wt strain were also performed using PacBio technology (GenBank accession number: CP183238). This assembly was used as a reference to identify mutations in related strains. Among the detected variants, only one mutation was unique to the NDmed *spsM::kan* (GM3248) strain and was not present in any of the other NDmed derivatives. This mutation was a thymine-to-adenine substitution at position 1222 of the *comP* coding sequence (comP:c.1222T>A). Bioinformatic analysis predicted that this substitution introduces a premature stop codon at the point of the mutation, likely resulting in a truncated, nonfunctional ComP protein. Notably, a secondary start codon located 15 bp downstream may allow expression of the C-terminal portion of ComP as a separate polypeptide. Together, these results led us to suspect that the altered macrocolony biofilm morphology observed in the NDmed *spsM::kan* (GM3248) strain might be due to a spontaneous *comP* mutation, rather than the *spsM* disruption alone.

Following this observation, two original stocks of the NDmed *spsM::kan* mutant (23), labeled GM3248 clone 1 and GM3248 clone 2, were tested for altered macrocolony morphology and screened for the presence of *comP* mutation by the research team at the Micalis Institute. The GM3248 clone 2 strain displayed the altered macrocolony morphology previously described by Sanchez-Vizuete et al. (23) and carried the same *comP* mutation as detected in the NDmed *spsM::kan* (GM3248) copy of the strain stored at Biotechnical Faculty, University of Ljubljana (Fig. S3). These findings demonstrate that the morphological phenotype originally attributed directly to *spsM* disruption in NDmed is likely linked to an undetected spontaneous mutation in *comP* and that disruption of *spsM* alone does not substantially alter biofilm morphology in either NDmed, P9_B1, or PS-216 genetic backgrounds.

## Complete deletion and spontaneous mutation of *comP* lead to altered colony biofilm morphology and impaired swarming motility

To determine whether the spontaneous *comP* mutation was responsible for the altered macrocolony biofilm morphology observed in the NDmed *spsM::kan* (GM3248) strain, we first constructed a complemented strain, NDmed *spsM::kan* (GM3248) *amyE::spsM* (MP). As expected, restoring *spsM* did not revert the morphology to wild type (Fig. 5A), supporting our hypothesis that the phenotype was not caused by disruption of *spsM*. We then introduced a copy of *comP* under IPTG-inducible control into the same strain (NDmed *spsM::kan* (GM3248) *amyE::comP*). Upon IPTG induction, the strain recovered wild-type morphology, confirming that the spontaneous *comP* mutation was indeed responsible for the altered biofilm phenotype. To test whether the premature stop codon in *comP* (comP:c.1222T>A) has the same effect as a complete gene deletion, we introduced *comP::ery* deletion in both NDmed *spsM::kan* (MP) and NDmed *spsM::kan* (GM3248). Both deletion strains showed identical macrocolony morphologies, which differed from their respective parental strains (Fig. 5A). This further supports that the NDmed *spsM::kan* (GM3248) strain carries a nonfunctional *comP* allele due to the spontaneous mutation. Other strains with *comP::ery* mutation were also constructed and tested on different types of media, and all had the same morphology, which was different from the *comP* spontaneous mutation comP:c.1222T>A (Fig. S4). Together, these results indicate that the premature stop codon in *comP* alters macrocolony biofilm morphology, but its effect is distinct from that of a complete *comP* deletion.

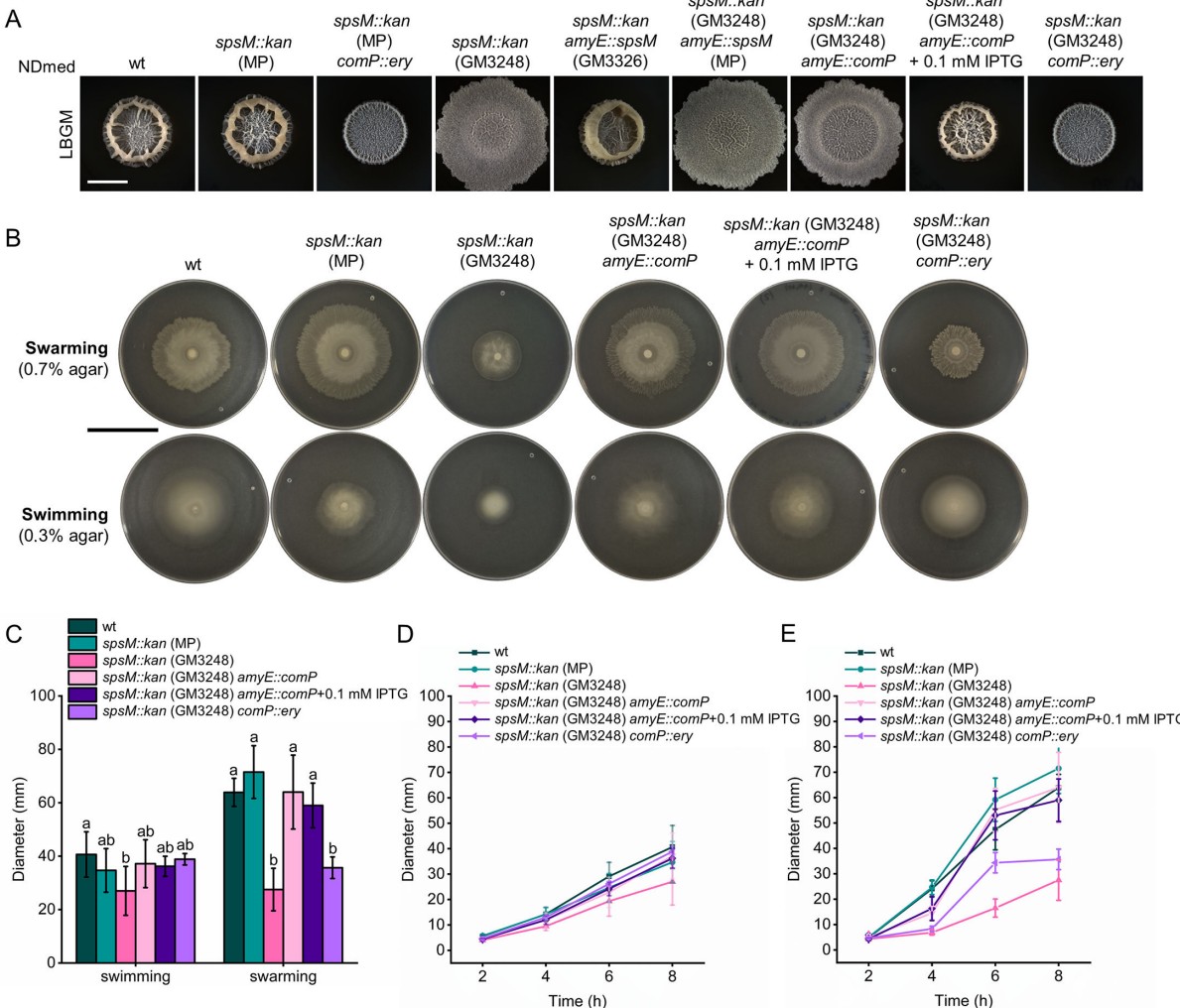

**FIG 5** Impact of distinct *comP* mutations on biofilm morphology and motility. All strain labels refer to NDmed derivatives. The strains labeled as wt, *spsM::kan* (GM3248), and *spsM::kan amyE::spsM* (GM3326) were previously described by Sanchez-Vizuete et al. (23). Strains marked with (MP) were constructed in this study. Images shown are representative of multiple independent experiments. (A) Images of macrocolony biofilms of strains with and without an active *comP*. Macrocolony biofilms were grown on solid LBGM medium and imaged after 48 h of incubation at 30°C. White scale bar: 5 mm. (B) Images of swarming and swimming motility of strains with and without an active *comP*. Plates were imaged after 8 h of incubation at 37°C. Plates were imaged against a black background so that bacterial colonization zones appear white, and uncolonized agar appears dark. Swimming assays were performed on LB with 0.3% agar, and swarming assays on LB with 0.7% agar. Black scale bar: 5 cm. (C) Quantification of swimming and swarming diameters after 8 h at 37°C. Columns represent mean diameters (*n* = 6), and error bars represent standard deviation. Statistical analysis was performed using one-way ANOVA followed by Tukey's Honest Significant Difference (HSD) test. Different letters above error bars indicate statistically significant differences between strains separately for swimming and swarming (*P* < 0.05). (D and E) Dynamics of swarming (C) and swimming (D) motility over an 8-hour period, with measurements taken at 2 h, 4 h, 6 h, and 8 h.

The *comP* gene is part of the ComQXPA quorum-sensing system in *B. subtilis* (65, 66) and is involved in expression of genes involved in surfactin production (67). Surfactin has been shown to enhance swarming motility in *B. subtilis* (68, 69). To further confirm the impact of the spontaneous *comP* mutation in the NDmed *spsM::kan* (GM3248) strain, we compared motility phenotypes of wild-type and mutant strains on LB medium containing 0.3% agar (for swimming) and 0.7% agar (for swarming). A significant reduction in swimming motility was observed only in the NDmed *spsM::kan* (GM3248) strain compared to the wild type, but not compared to the rest of the tested strains (Fig. 5B and C). The effect of the *comP* mutation was, however, more pronounced in swarming assays. Both the NDmed *spsM::kan* (GM3248) strain carrying the spontaneous *comP* mutation and the NDmed *spsM::kan* (GM3248) *comP::ery* deletion mutant exhibited

reduced swarming motility compared to strains with an intact *comP* gene (Fig. 5B and C). This reduction in motility was consistent across all measured time points, as shown in Fig. 5D and E.

It has previously been shown that surfactin can be visualized as an expanding ring ahead of the swarming front, visible under reflected light (70). In line with this observation, strains lacking an active *comP* gene (NDmed *spsM::kan* (GM3248) and NDmed *spsM::kan* (GM3248) *comP::ery*) did not produce a visible surfactin ring during swarming. In contrast, the ring reappeared in the NDmed spsM::kan (GM3248) *amyE::comP* strain, where *comP* was complemented by an additional copy. Notably, this complementation restored both swimming and swarming motility, as well as surfactin production, even in the absence of IPTG, likely due to leaky promoter expression.

Interestingly, although the premature stop codon mutation (comP:c.1222T>A) and the complete deletion of *comP* produced similar swarming defects and lack of a surfactin ring, they resulted in distinct macrocolony biofilm morphologies (Fig. 5A), suggesting that these two mutations have different downstream effects on surface-associated phenotypes. Since bioinformatic analysis of the comP:c.1222T>A mutation predicts a secondary start codon located 15 bp downstream of the mutation, it is possible that both the N-terminal and C-terminal portions of ComP are separately translated and affect biofilm morphology in a distinct way. Another plausible explanation is that the two alleles differentially affect expression of *comA*, the gene immediately downstream of *comP*, which could, in turn, alter regulation and contribute to the observed phenotypic differences.

## The *spsM::kan* permanent gene disruption increases mutation rates, which results in mutations affecting motility and biofilm morphology

In strains carrying the *spsM::kan* mutation, two spontaneous mutations were identified: a deletion in *swrA* (c.26delT) in the P9_B1 fluffy strain and a substitution in *comP* (c.1222T>A) in the NDmed *spsM::kan* (GM3248) strain. These findings led us to suspect that disruption of *spsM* may accelerate the appearance of additional spontaneous mutations. To investigate this further, we first screened selected strains for spontaneous appearance of peripheral outgrowths. These outgrowths were restreaked and assessed for altered colony morphology (Fig. S5A). Across four independent experiments, we screened a total of 140 colonies per strain, providing sufficient sampling depth to detect spontaneously emerging variants. Because minor differences in growth conditions seemed to influence the frequency of outgrowths between experiments, the data were not pooled together and were not subjected to statistical comparison. Across four independent experiments (140 colonies per strain), we observed 0/140 outgrowths in P9_B1 wt, 23/140 in *spsM::kan* (10 mut-unk; 13 mut-swrA), 11/140 in *spsM::kan amyE::spsM* (all mut-swrA), and 1/140 in Lys SPβ (mut-comP). Counts for each experiment are provided in Fig. S5.

Aside from the flat colony morphotype similar to "fluffy," we also observed an emergence of wider flat (similar to NDmed *spsM::kan* (GM3248) *comP* mutant) and glossy colony types (Fig. 6). A subset of representative strains from each morphotype was subjected to sequencing of *swrA* and *comP* loci. It turned out that all sequenced isolates with a shared morphology carried the same mutation pattern within *swrA* and *comP*, allowing their classification based on phenotype. Strains labeled as mut-swrA displayed a flat and expanded colony morphology and carried the *swrA* c.26delT deletion. Mut-comP strain showed a wider, flatter morphology with reduced motility and carried a frameshift mutation in *comP*. Mut-unk strains exhibited a glossy colony appearance and altered swarming dynamics, although the exact mutation responsible was not identified (Fig. 6; Fig. S5).

Despite potential variability between experiments, a consistent qualitative pattern emerged. No mutants were identified in the 140 colonies of the P9_B1 wt strain across all experiments. In contrast, spontaneous mutants described above repeatedly appeared in *spsM::kan* and *spsM::kan amyE::spsM* strains. Interestingly, the one *comP* mutation

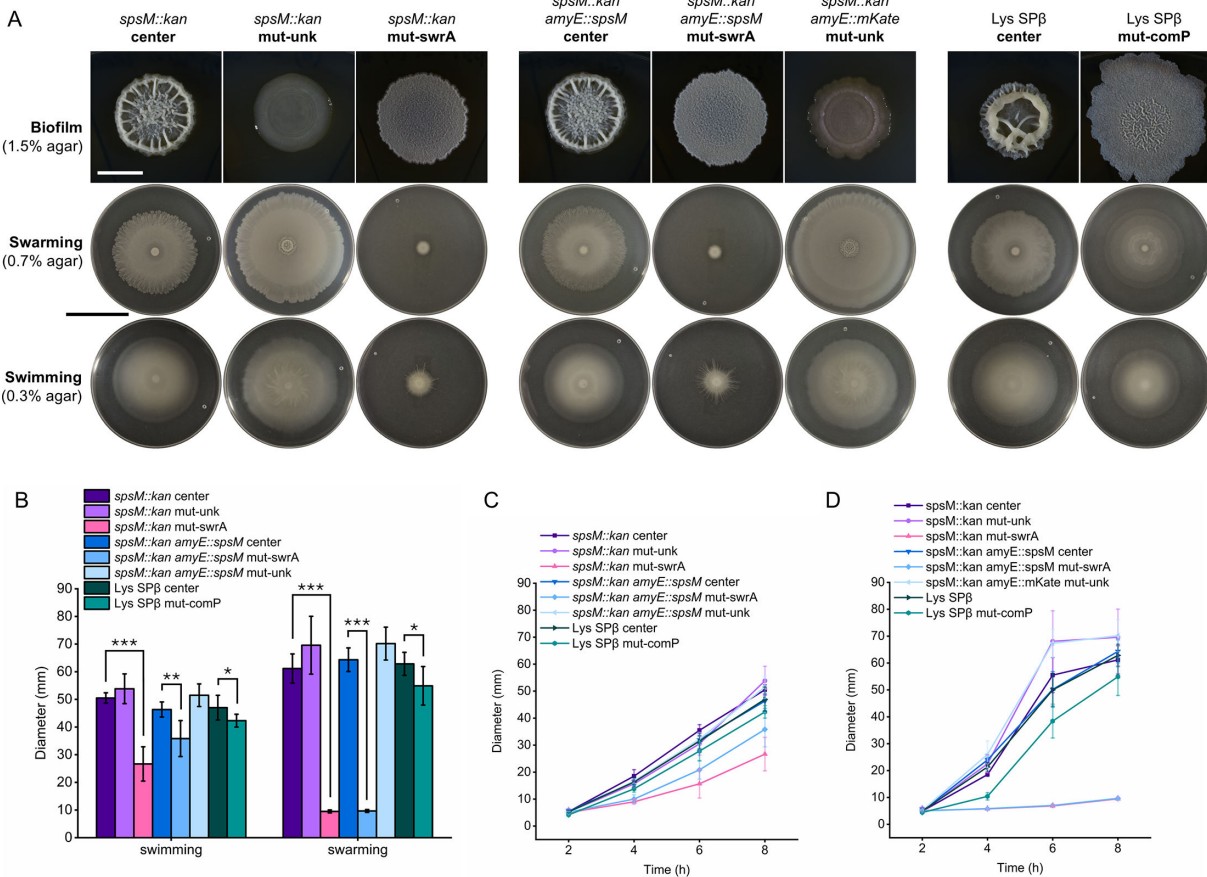

**FIG 6** Impact of detected spontaneous mutations on biofilm morphology and motility. All strain labels refer to P9_B1 derivatives. (A) Representative images of macrocolony biofilms, swarming motility, and swimming motility for spontaneous mutants. Strains were derived from the center (labeled as "center") and outgrowth (labeled as "mut-") regions of macrocolonies grown for 48 h at 30°C on solid LB medium. Mutants labeled as mut-swrA are associated with the *swrA* c.26delT deletion, mutant mut-comP carries the *comP* c.1115delT deletion, and mut-unk refers to strains with unknown mutations. Macrocolony biofilms were grown on LBGM agar and imaged after 48 h at 30°C. Swarming and swimming assays were performed on LB containing 0.7% and 0.3% agar, respectively, and imaged after 8 h of incubation at 37°C. Plates were imaged against a black background so that bacterial colonization zones appear white, and uncolonized agar appears dark. White scale bar: 5 mm. Black scale bar: 5 cm. (B) Quantification of swimming and swarming diameters after 8 h at 37°C. Columns represent mean diameters ($n = 6$), and error bars represent standard deviation. Statistical analysis was performed using one-way ANOVA followed by Tukey's Honest Significant Difference (HSD) test, performed separately for each parental strain (*$P < 0.05$, **$P < 0.01$, ***$P < 0.001$). (C and D) Dynamics of swarming (C) and swimming (D) motility over an 8-hour period, with measurements taken at 2 h, 4 h, 6 h, and 8 h.

(c.1115delT) emerged in the P9_B1 Lys SPβ strain. Although its macrocolony morphotype resembled the NDmed *spsM::kan* (GM3248) strain carrying a different spontaneous *comP* mutation (c.1222T>A), the two strains displayed distinct swarm patterns. This suggests that different mutations in *comP* can lead to distinct effects on motility, even when overall colony morphology appears similar (Fig. 6).

As additional controls, we tested 40 colonies of P9_B1 *amyE::mKate* and *spsM::kan amyE::mKate* strains. A single mutant appeared in the *spsM::kan amyE::mKate* strain, displaying the same glossy phenotype as the mut-unk group (Fig. 6; Fig. S5). No mutations were observed in the *amyE::mKate* control. These results indicate that disruption of the *amyE* locus does not promote spontaneous variant formation. Instead, the increased diversification observed in *spsM::kan* and *spsM::kan amyE::spsM* strains shows that complementation does not restore the phenotypic stability of the *spsM::kan* background.

To determine whether the observed differences in diversification reflected variation in population size rather than differences in mutation supply, we quantified CFU/mL from

macrocolonies grown under the same conditions used for the diversification assay. Total cell numbers were comparable across P9_B1 wt, *spsM::kan*, *spsM::kan amyE::spsM*, and Lys SPβ strains (Fig. S6), indicating that the higher frequency of peripheral outgrowths in the *spsM::kan* mutant cannot be explained by bigger population size.

To determine whether mutation rate differences help explain the observed diversification patterns, we performed a rifampicin fluctuation assay. To establish whether elevated mutation rates were resulting from the loss of *spsM* function or were due to the specific insertional disruption created by *spsM::kan*, we constructed an additional markerless Δ*spsM* strain. We also included the BKE17040 *mutS::ery* strain as a positive control mutator to validate that our assay and analysis pipeline reliably detect increased mutation rates.

We first showed that planktonic cultures used in the assay displayed indistinguishable CFU/mL values (Fig. S6). As expected, the *mutS::ery* control exhibited a strongly elevated mutation rate ($2.06 \times 10^{-7}$), confirming that the protocol robustly captures high-mutation-rate phenotypes under our conditions. The mutation rate estimations and two-sample likelihood ratio test revealed that the P9_B1 *spsM::kan* strain exhibits a significantly elevated mutation rate compared to P9_B1 wt ($2.15 \times 10^{-9}$ vs $4.90 \times 10^{-10}$; $P = 0.003$) (Fig. 7). The P9_B1 SPβ lysogen showed only a modest upward trend relative to the P9_B1 wt ($1.08 \times 10^{-9}$ vs $4.90 \times 10^{-10}$), but this difference was not statistically significant ($P = 0.076$). Complementation of *spsM* at the *amyE* locus yielded an intermediate mutation rate ($9.20 \times 10^{-10}$) that was not statistically different from either P9_B1 wt strain ($P = 0.085$) or P9_B1 *spsM::kan* strain ($P = 0.157$), demonstrating that restoring *spsM* in *trans* did not fully reverse the mutator phenotype. Importantly, the markerless Δ*spsM* strain did not differ from P9_B1 wt ($8.03 \times 10^{-10}$ vs $4.90 \times 10^{-10}$; $P = 0.428$), while remaining significantly lower than *spsM::kan* ($P = 0.022$), indicating that the elevated mutation rate is linked to the *spsM::kan* insertion rather than *spsM* loss. These data suggest a cis-acting effect of the kanamycin resistance cassette at the native *spsM* locus, rather than a phenotype attributable to loss of SpsM protein. As an additional transparency note, sequencing of the P9_B1 *spsM::kan* strain also identified a unique nonsynonymous variant in *ywdH* that was absent from all other P9_B1 and NDmed genomes sequenced in this study. While *ywdH* remains functionally uncharacterized and the change is a single amino-acid substitution, we cannot exclude that it contributes to the elevated mutation rate alongside the cassette insertion.

## Spontaneous point mutation in *swrA* provides a competitive advantage to strains with disrupted *spsM* in laboratory biofilm-promoting conditions

Among the spontaneous mutations identified in our screening, the most frequent was a single-base deletion in *swrA* (c.26delT). This mutation consistently occurred in the P9_B1 *spsM::kan* strain at the same position and resulted in loss of swarming motility and a less complex biofilm morphology. Given that reduced biofilm complexity is commonly observed in laboratory strains of *B. subtilis* (72), we investigated whether the *swrA* c.26delT mutation, or a complete deletion of the gene (*swrA::tet*), might provide a fitness advantage in *spsM*-negative genetic background under laboratory biofilm-promoting conditions. To assess this, we performed competition assays on solid LBGM medium using 1:1 mixtures of fluorescently labeled strains. Cultures were incubated for 48 h at 30°C, and fluorescence was quantified by measuring integrated density from colony images (Fig. 8). Because GFP and mKate fluorescent signals varied in intensity, monocultures were used to normalize the fluorescence values of cocultures (Fig. 8A).

Fluorescent labels were first confirmed to have only a minimal impact on competitive fitness by comparing isogenic strains expressing either GFP or mKate. Next, the P9_B1 *spsM::kan* strain was mixed with the spontaneous mutant P9_B1 fluffy, which carries the *swrA* c.26delT mutation. In both labeling configurations, the fluffy strain showed a modest increase in relative abundance compared to the isogenic control, suggesting a slight competitive advantage (Fig. 8B and C). When the *spsM::kan* strain was competed against the *spsM::kan swrA::tet* deletion strain, the effect was more pronounced. In both

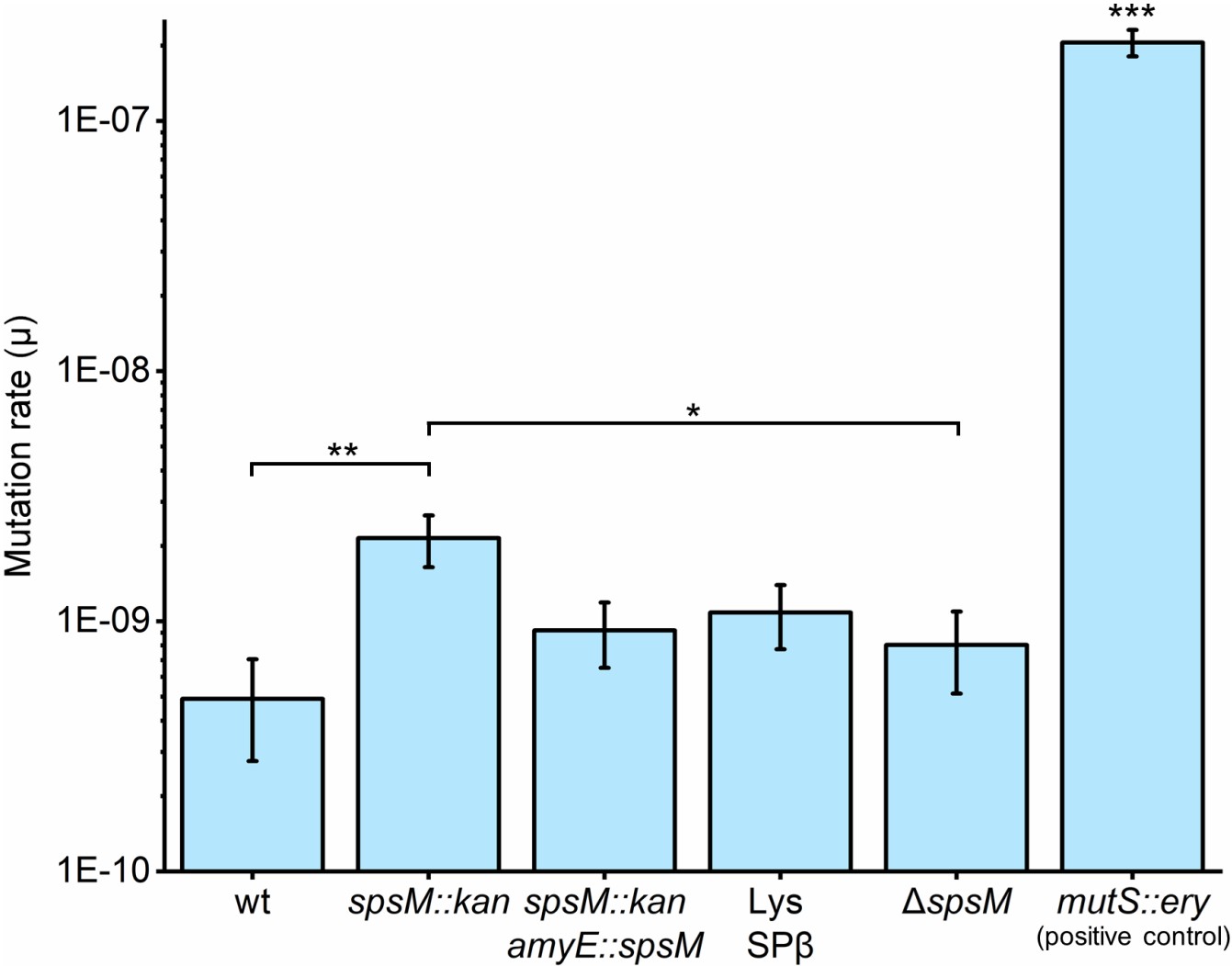

**FIG 7** Mutation rate estimates for *B. subtilis* P9_B1 derivatives. Mutation rates are estimated from rifampicin fluctuation assays. Columns represent mean mutation rates ($n = 84$), and error bars indicate standard deviation. Mutation parameters were determined using the RStudio *flan* package. Final population sizes measured for each culture were used to convert mutation parameters to per-cell mutation rates. Pairwise comparisons between strains were performed using *flan's* two-sample likelihood ratio test. Statistical significance is indicated as *$P < 0.05$, **$P < 0.005$, ***$P < 0.001$. Lastly, we performed a phylogenetic comparison of the two natural isolates used throughout this study, P9_B1 and NDmed (Fig. S7). The analysis showed that P9_B1 and NDmed are very closely related at the whole-genome level, indicating that the diversification phenotypes we observe are not restricted to a single, unusually divergent background. Multi-Locus Sequence Analysis (MLST) (59) clearly shows very high similarity between P9_B1 and NDmed in the context of other closely related *Bacillus* sp. strains. In addition, ANI (average nucleotide identity) between the two strains is 99.957%, which is very high, even considering within-species ANI values (61, 71). Consistent with this, spontaneous mutations were detected in macrocolonies derived from *spsM::kan* strains in both genetic backgrounds, supporting the interpretation that the same *spsM::kan*-associated mechanism promotes the rapid emergence of secondary mutations in each strain.

label combinations, the strain with the complete *swrA* deletion consistently outcompeted the strain with an intact *swrA* gene, indicating a clear fitness advantage under conditions with enhanced biofilm production. Interestingly, competition between the *spsM::kan swrA::tet* strain and the fluffy mutant again showed a competitive advantage for the deletion strain, even though both strains lacked a functional *swrA* gene. This mirrors the pattern observed for *comP*, where complete deletion had different phenotypic consequences compared to spontaneous point mutations.

In contrast, the same competitive advantage was not observed on LB medium alone (Fig. S8), indicating that the benefit of *swrA* deletion is condition dependent. These results suggest that loss of *swrA* function enhances fitness specifically under

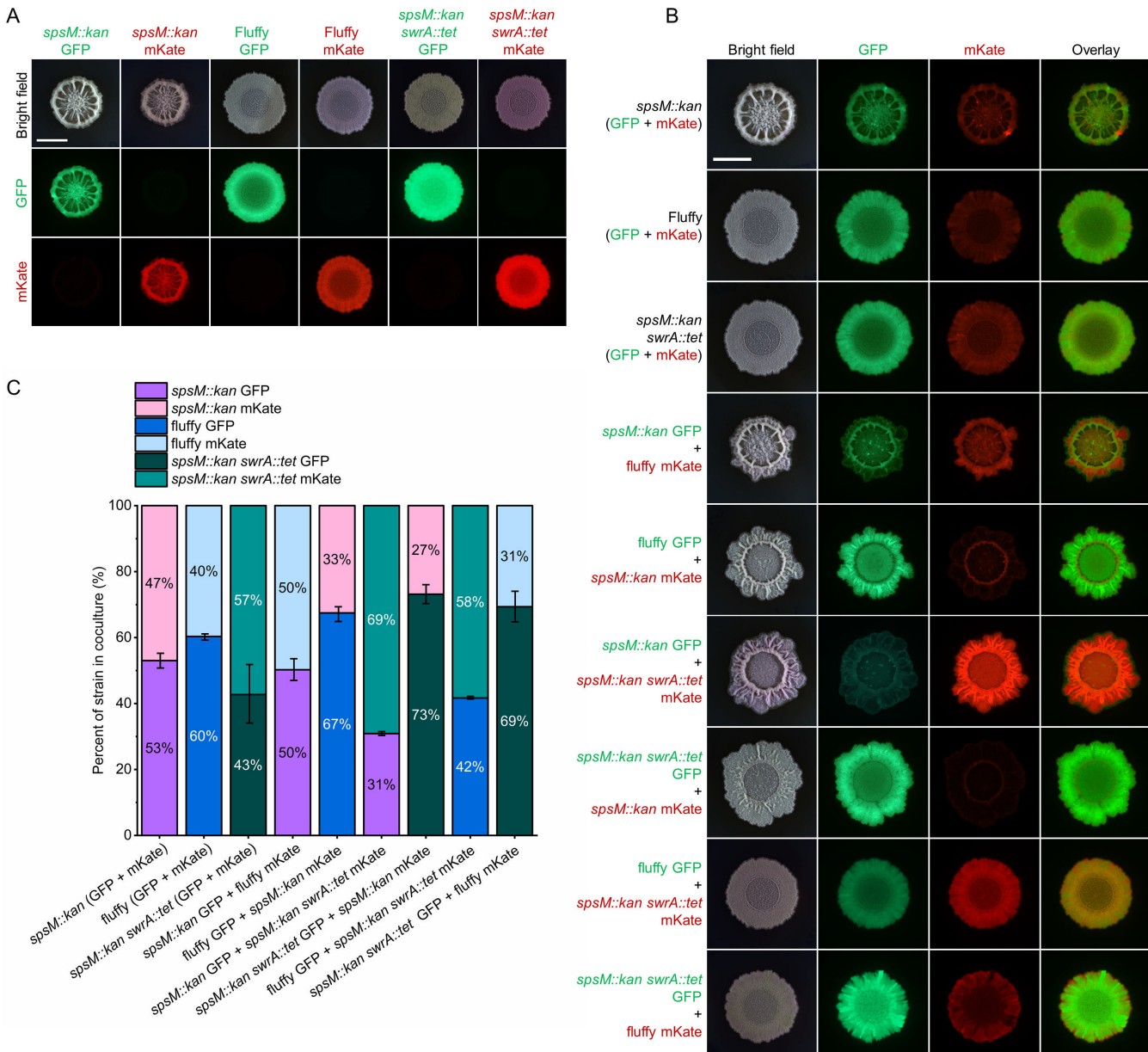

**FIG 8** Competitive coculture assays comparing *swrA*-active and *swrA*-inactive strains in biofilm conditions. All strain labels refer to P9_B1 derivatives. The fluffy strain refers to the *spsM::kan* strain carrying the spontaneous swrA c.26delT mutation. (A and B) Representative macrocolony biofilms of monocultures (A) and 1:1 cocultures (B) grown on solid LBGM medium and incubated for 48 h at 30°C. Fluorescently labeled strains (GFP and mKate) were used to distinguish genotypes within each mixture. Scale bar: 5 mm. (C) Relative abundance of each strain in cocultures presented as a 100% stacked bar plot (*n* = 4). Fluorescence intensity was measured from macrocolony images using Fiji (ImageJ), and values were normalized using monoculture fluorescence to account for differences between GFP and mKate signal strength. Each bar shows the proportion of each strain in the coculture, reflecting its competitive fitness in biofilm-promoting conditions.

biofilm-promoting conditions on LBGM, likely by reducing the energetic cost of motility or regulatory interference. This may explain why the *swrA* c.26delT mutation was frequently recovered in our screen, as spontaneous mutations that promote growth under biofilm-favoring conditions are likely to be rapidly selected.

## DISCUSSION

The initial objective of our study was to examine the functional consequences of prophage integration at the *B. subtilis* locus *spsM*, previously implicated in biofilm

formation. The unexpected discovery of frequent secondary loss-of-function mutations in *spsM::kan* strains initially suggested that *spsM* might contribute to maintaining genomic stability or that its disruption imposes strong selection for reduced motility and biofilm formation. To distinguish an effect of *spsM* loss from an effect specific to cassette insertion at the native locus, we generated a markerless Δ*spsM* mutant as a critical control. Unlike *spsM::kan*, the markerless Δ*spsM* did not show an increased mutation rate relative to wild type, demonstrating that elevated mutation supply is not a general consequence of *spsM* inactivation. Complementation of *spsM* at an ectopic locus also did not restore the wild-type mutation rate, indicating that the elevated mutational burden is unlikely to result from loss of SpsM function. Instead, the kanamycin resistance cassette at the native *spsM* locus is a more plausible driver of the mutator phenotype.

Our rifampicin fluctuation assay showed an elevated per-cell mutation rate in the *spsM::kan* mutant strain. Measurements of CFU/mL from macrocolonies and liquid planktonic cultures indicate comparable population sizes across all genetic backgrounds, indicating that the higher frequency of visible variants in the *spsM::kan* strain reflects increased mutation supply rather than larger population size. Increased mutation rate in this mutant could arise from a cis-acting effect of the kanamycin cassette that affects expression or regulation of genes neighboring *spsM*. In the *B. subtilis* P9_B1 background, *msrA* and *msrB* form a single operon immediately downstream of *spsM*, and their expression is strongly integrated into the Spx-dependent oxidative stress response (73). MSR enzymes are central components of oxidative-stress defense, and loss or reduction of MsrA/B activity leads to accumulation of oxidized proteins and increased sensitivity to reactive oxygen species (74–77). Work in multiple bacterial species has shown that oxidative damage can elevate spontaneous mutation rates and accelerate acquisition of resistant or adapted genotypes (78–81). A modest, chronic increase in oxidative damage could therefore raise the overall spontaneous mutation rate (as estimated by rifampicin resistance) in the *spsM::kan* background, without requiring any specific hypermutable neighborhood at *swrA* or *comP*. Consistent with this, resistance-cassette insertions have been shown to exert polar effects on downstream transcription in *B. subtilis*, and systematic analyses of cassette-based deletion alleles indicate that antibiotic-cassette replacements can frequently alter expression of neighboring operon genes (82, 83). In addition to the downstream *msrAB* operon, the *spsM* locus is flanked upstream by *phyC* and the adjacent *cgeAB* operon, any of which could also be subject to cassette-associated polar effects or local regulatory perturbations in the *spsM::kan* background. Definitively resolving the mechanistic basis of the mutator phenotype would require direct measurement of transcriptional consequences in these neighboring genes (e.g., RT-qPCR) or reconstruction of targeted perturbations of these loci, which is beyond the scope of the present study. Although our data indicate that the elevated frequency of rifampicin-resistant colonies in the *spsM::kan* background is most consistent with an increased mutation supply, we acknowledge that a parallel effect on the strength of selection cannot be ruled out. Antibiotic resistance cassettes can exert polar or off-target effects on neighboring genes, and such regulatory perturbations can modify antibiotic susceptibility phenotypes, thereby influencing the effective selection environment, as shown for cassette-associated repression and downstream transcriptional changes in other bacterial systems (82, 84).

Importantly, SPβ lysogeny, despite interrupting *spsM*, did not reproduce the *spsM::kan*-associated mutator phenotype under our assay conditions. Although we observed a visible variant morphotype in the SPβ lysogen, the rifampicin fluctuation assay did not detect a statistically significant increase in mutation rate relative to wild type. This dissociation suggests that prophage integration at *spsM* and cassette insertion at the same locus have distinct genetic consequences, and it underscores that interruption of *spsM* is not, by itself, predictive of increased mutation supply. Active lysogeny remains a plausible contributor to phenotypic buffering at the *spsM* locus during specific developmental states (20, 24). Accordingly, our data do not support an interpretation in which SPβ integration at *spsM* adaptively manipulates biofilm formation. Any ecological

scenarios linking *spsM* targeting to transmission remain speculative. In addition, precise developmental excision during active lysogeny restores *spsM* integrity and should not be equated with DNA-damage–induced prophage induction (e.g., mitomycin C), which typically initiates lytic development and host lysis.

Higher mutation supply, combined with strong selection at colony edges, is sufficient to explain the recurrent appearance of regulatory gene mutations we detected. In spatially expanding colonies, evolutionary dynamics are often dominated by the growth frontier, where advantageous lineages can expand from the frontier and produce sector-like expansions, reflecting strong selection at colony edges (85, 86). Under our macrocolony growth conditions, mutations that disrupt key regulatory genes with large phenotypic effects, such as the motility and biofilm regulators *swrA* and *comP*, are repeatedly generated and can expand along colony edges, consistent with spatial structure turning surface growth into competition for space and selecting for increased surface spreading (5, 87). In this setting, the elevated mutation rate in *spsM::kan* would increase the supply of *swrA/comP* variants, and selection at the expanding edge would preferentially amplify lineages with advantageous shifts in motility–biofilm regulation (88). Genome-wide mutation-accumulation work in *B. subtilis* shows strong context dependence of site-specific mutation rates, but does not single out the *swrA* or *comP* regions as exceptional hotspots (89). Instead, our observations are consistent with these loci being functionally accessible targets under laboratory selection.

Loss-of-function mutations in key regulators are a common feature of microbial adaptation to laboratory conditions (a process referred to as domestication) and often target regulatory genes that govern energy-intensive traits, such as motility and biofilm formation (13, 16, 90–95). Similar patterns have also been reported in experimental evolution studies with *Pseudomonas*, *Escherichia*, and *Bacillus*, where adaptation to favorable laboratory conditions frequently results in loss-of-function mutations that downregulate redundant or costly processes (16, 91, 96–98). Along these lines, in *B. subtilis*, mutations in *comP* and *swrA* affect multiple interconnected phenotypes and often confer a selective advantage in nutrient-rich media by eliminating costly communal behaviors (16, 91, 99), such as collective motility, thereby conferring a fitness advantage. Our repeated recovery of *swrA* and *comP* variants, together with the potential fitness benefit of *swrA* loss under biofilm-promoting conditions, is consistent with a domestication-like process in which selection favors reduced investment in costly social behaviors in nutrient-rich laboratory environments. We quantified competition consequences for *swrA* loss under LBGM macrocolony conditions. The fitness effects of *comP* variants and of the different *spsM* disruption backgrounds were not systematically quantified and could contribute via epistasis.

The *swrA* frameshift mutations are a hallmark of domesticated, non-swarming *B. subtilis* strains (32, 63), and the ComQXPA locus, including the sensor domain of ComP, is highly polymorphic and subject to diversifying selection in natural populations (95, 100, 101). Together, this supports a model in which increased genome-wide mutation supply in the *spsM::kan* background, combined with edge selection on regulatory trade-offs, drives the recurrent emergence of *swrA* and *comP* mutants without requiring that these loci reside in globally hypermutable regions. Additionally, phylogenetic comparison based on whole-genome sequences indicates that NDmed and P9_B1 are closely related, supporting that the same *spsM::kan*-associated dynamics can manifest across closely related natural backgrounds. Epistatic interactions may still modulate how readily secondary mutants become visible, because the physiological consequences of cassette insertion versus SPβ integration could shift the fitness effects of newly arising *swrA/comP* variants, an effect we did not quantify across backgrounds. Consistent with this, secondary mutations were detected in *spsM::kan*-derived lineages from both strains.

Mutations in *swrA* consistently led to a loss of swarming motility, aligning with its known function as a regulator of flagellar assembly regulation (63, 64, 68). Notably, reversible mutations in *swrA*, such as an A–T base pair insertion, have previously been described as arising via slipped-strand mispairing during replication (62, 102). The same

frameshift mutation is fixed in lab strains like *B. subtilis* 168, which lack swarming capability (32, 103). Beyond motility loss, we observed *swrA* mutations also altered biofilm morphology, likely due to downstream regulatory effects on exopolysaccharide and γ-poly-DL-glutamic acid (γ-PGA) synthesis, which are critical for robust biofilm formation (63, 104).

Similarly, mutations in *comP* impaired swarming motility, likely by abolishing surfactin production, which has been shown to be a key surfactant required for surface-spreading motility (68). This is consistent with prior studies showing that ComP regulates surfactin and iturin biosynthesis through ComA phosphorylation, directly influencing both colony expansion and biofilm integrity (67, 95, 105, 106). Our observations of surfactant-deficient colonies and altered morphology in *comP* mutants align with previously reported phenotypes in *B. subtilis* morphology (32, 63, 70, 95). In addition to surfactin production, ComP-mediated signaling modulates other regulators, such as DegU, which further orchestrate motility and biofilm pathways (67, 107).

Taken together, our results refine the interpretation of prior work on the role of *spsM* in biofilm formation (23). Using whole-genome sequencing and reconstruction controls, we show that the prominent NDmed biofilm defect previously attributed to *spsM::kan* is instead explained by an undetected secondary mutation in comP. Our results indicate that disruption of *spsM* alone (without secondary loss-of-function mutations) did not significantly alter host biofilm morphology or motility phenotypes. This observation was true for both the permanent disruption mutant (*spsM::kan*) and the SPβ lysogen, in which *spsM* is interrupted by prophage integration. These findings suggest a more refined understanding of *spsM* function and may call for a re-evaluation of the earlier hypothesis that *spsM* disruption promotes phage dissemination by restructuring biofilm spatial organization (23, 27).

More broadly, our findings highlight how prophage biology and experimental genetics intersect at chromosomal integration loci. Prophage carriage can reshape host phenotypes and social interactions in *B. subtilis* (17, 21, 108), but our results show that the experimental route used to modify an integration locus can also have major and unexpected consequences. Specifically, cassette insertion at *spsM* increased mutation supply and accelerated recovery of recurrent regulatory mutants, whereas markerless deletion and SPβ lysogeny did not reproduce the same mutation-rate phenotype. This distinction underlines the importance of genetic engineering strategy, markerless controls, and routine genome verification when attributing phenotypes in mutant strains to the intended genetic change, particularly for traits that are sensitive to domestication and selection in laboratory environments.

## ACKNOWLEDGMENTS

M.P. and A.D. were supported by the Slovenian Agency for Research and Innovation (ARIS) grant no. P4-0116 and the European Union (ERC, PHAGECONTROL, 101041421). The views and opinions expressed are, however, those of the authors only and do not necessarily reflect those of the European Union or the European Research Council Executive Agency. Neither the European Union nor the granting authority can be held responsible for them. I.D. was supported by ARIS grants P4-0116 and J1-3021. T.A. was supported by an ARIS grant P4-0097. Y.D. and R.B. were supported by supported by INRAE. The stereomicroscopy experiments were supported by funding from ARIS to Infrastructural Centre Microscopy of Biological Samples (MRIC UL, I0-0022-0481-08), at Biotechnical Faculty, University of Ljubljana, Slovenia.

We thank Polonca Štefanič for advice regarding strain relatedness analysis, phylogeny, and ANI calculations.

## AUTHOR AFFILIATIONS

[1]Department of Microbiology, Biotechnical Faculty, University of Ljubljana, Ljubljana, Slovenia

<sup>2</sup>Université Paris-Saclay, INRAE, AgroParisTech, Micalis Institute, Gif-sur-Yvette, France

## AUTHOR ORCIDs

Romain Briandet http://orcid.org/0000-0002-8123-3492
Anna Dragoš http://orcid.org/0000-0003-4136-986X

## FUNDING

| Funder | Grant(s) | Author(s) |
| --- | --- | --- |
| Slovenian Agency for Research and Innovation | P4-0116 | Maja Popović |
| | | Anna Dragoš |
| | | Iztok Dogša |
| HORIZON EUROPE European Research Council | 101041421 | Maja Popović |
| | | Anna Dragoš |
| Slovenian Agency for Research and Innovation | P4-0097 | Tomaž Accetto |
| Slovenian Agency for Research and Innovation | J1-3021 | Iztok Dogša |
| Institut National de Recherche pour l'Agriculture, l'Alimentation et l'Environnement | | Romain Briandet |
| | | Yasmine Dergham |

## AUTHOR CONTRIBUTIONS

Maja Popović, Conceptualization, Data curation, Formal analysis, Investigation, Methodology, Validation, Visualization, Writing – original draft, Writing – review and editing | Tomaž Accetto, Formal analysis, Methodology, Supervision, Validation, Writing – review and editing | Yasmine Dergham, Formal analysis, Investigation, Validation, Visualization, Writing – review and editing | Romain Briandet, Formal analysis, Investigation, Resources, Supervision, Validation, Writing – review and editing | Iztok Dogša, Conceptualization, Formal analysis, Methodology, Software, Supervision, Validation, Writing – review and editing | Anna Dragoš, Conceptualization, Data curation, Funding acquisition, Project administration, Resources, Supervision, Writing – original draft, Writing – review and editing

## ADDITIONAL FILES

The following material is available online.

### Supplemental Material

**Supplemental material (Spectrum02504-25-s0001.docx).** Fig. S1 to S8.

### Open Peer Review

**PEER REVIEW HISTORY (review-history.pdf).** An accounting of the reviewer comments and feedback.

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
