## [Reviewer comments · Microbiology Spectrum]

Microbiology Spectrum

Disruption of bacteriophage integration site promotes rapid diversification of multicellular traits in *Bacillus subtilis*

Maja Popović, Tomaz Accetto, Yasmine Dergham, Romain briandet, Iztok Dogša, and Anna Dragoš

Corresponding Author(s): Anna Dragoš, Univerza v Ljubljani Biotehniška fakulteta

Review Timeline:

Submission Date:	August 20, 2025
Editorial Decision:	October 7, 2025
Revision Received:	February 10, 2026
Editorial Decision:	March 2, 2026
Revision Received:	March 12, 2026
Accepted:	March 16, 2026

Editor: Olaya Rendueles

Reviewer(s): Disclosure of reviewer identity is with reference to reviewer comments included in decision letter(s). The following individuals involved in review of your submission have agreed to reveal their identity: Avigdor Eldar (Reviewer #1); Sébastien Wielgoss (Reviewer #2)

Transaction Report:

DOI: <https://doi.org/10.1128/spectrum.02504-25>

Re: Spectrum02504-25 (Disruption of bacteriophage integration site promotes rapid diversification of multicellular traits in *Bacillus subtilis*)

Dear Dr. Anna Dragoš:

Thank you for allowing us to review your work. I am really sorry for the delay, I was waiting for one last review. Both reviewers are convinced of the interest and high potential of this work. However, there are some concerns concerning the way the manuscript is written and both suggest modifications on how the story is framed. I would encourage significant rewriting. Below you will find instructions from the Spectrum editorial office, and the reviewer comments.

Revision Guidelines

Sincerely,
Olaya Rendueles
Editor
Microbiology Spectrum

Reviewer #1 (Comments for the Author):

The manuscript by Popović et al studies the effect of either deletion of, or integration into, the *spsM* gene, where SPbeta phage integrates in the lab strain. To this aim, they mostly examine the effect of this deletion in two wild isolates. This is a follow-up paper to a previous work from the same lab (Sanchez-Vizuetes et al., 2015) which identified changes in biofilm phenotype upon

deletion of this gene in one of the wild isolates.

The work has several main results:

1. The previous work was based on an unfortunate artifact - the *spsM* (*ypqP*) mutant strain examined by Sanchez-Vizueté had an additional mutation in the *comP* gene. This mutation led to the biofilm phenotype led to the perturbed biofilm phenotype and not the *spsM* mutation. The author has previously done complementation assays, but apparently the *comP* mutation did not appear in the *spsM* complementation strain. Therefore, complementation seemed to indicate a direct role of *spsM* in biofilm formation.
2. The *spsM* mutant accumulated additional mutations at the edge of the colony in either the *comP* and *swrA* genes (and in an additional unidentified locus). These mutations tend to have a strong (and different) biofilm phenotype, which is carefully characterized by the authors. It is shown that these mutations are selected under strong biofilm producing conditions.
3. Mutations in *comP* and *swrA* tend to form at a much higher rate on the *spsM::kan* mutant than on the wild-type. This is still true in the *spsM* complementation strain. The SPbeta lysogen of one of the wild isolates showed some mutant phenotypes, but at a much lower rate.

General:

It is to be commended that the authors alert the community to an unintentional mistake which occurred in a previous manuscript. The analysis of the biofilm phenotype of the *comP* and *swrA* mutant is also interesting, though I think that it generally matches what's expected from the deletions based on what is known about their role in strain 3610 - the model strain for biofilm formation. However, I find the main claim of the manuscript - that *spsM* is an evolutionary hotspot that influence genome integrity and adaptive potential - to be unsatisfactory and over-hyped. The most likely hypothesis in my mind is that the deletion marker leads to over- or under-expression of a neighboring gene, which puts a stronger selection pressure for compensating biofilm mutants. This is not really what one would call a hot spot for adaptive potential. Further work should be done to estimate the sources for the increased phenotypic diversification rate.

Major points:

1. The role of the *spsM::kan* mutation in supporting increased phenotypic (and genetic) diversity. The most likely source of the effect is the selection marker and its promoter. The authors should re-estimate the rate of diversification in a marker-less deletion of *spsM* - this is easy to do and would greatly help in distinguishing between a cis effect of the DNA sequence of the *spsM* locus and a trans effect of marker expression (most likely) on neighboring genes.

If the increased genetic diversity is specific to the *kan*-marked mutant, I would suggest studying the over expression or deletion (according to the presumed effect) of the genes neighbouring the *spsM* gene. In strain 168 these are the *msrAB* genes (Methionine sulfoxide reductases) or *ypoP* (a transcription factor of unknown function) on one side and the *phy* (phytase) gene or *cge* operon (spore maturation) on the other side (based on SubtiWiki) - both are not known to affect selection on biofilm, but they may have such an effect. Alternatively, these might be different genes in the wild isolates used by the authors.

If the authors find that the marker-less deletion still has the increased diversity effect, it probably suggests the presence of a regulatory region on that locus. I would therefore also recommend doing RT-PCR of the neighboring genes in different genetic backgrounds to verify an effect on their expression level.

2. The claim of increased phenotypic diversity in mutants and lysogen, compared to wild-type should be better quantified. Specifically, the claim that the lysogen led to the emergence of mutant in one out of 40 colonies while no mutants were observed in the wild-type is definitely not statistically significant. It is easy to estimate the rate of emergence of visible mutant phenotype in several hundreds of colonies of each type and do a proper statistical analysis.

3. The paper's writing is somewhat convoluted. I would have liked to understand that the previously associated biofilm phenotype of the *spsM* mutant was a mistake before reaching line 535. I think that the claims on being a "hot-spot" for diversity should be toned down even if they find a different mechanism underlying this process than what I suspect it is.

Minor:

Line 189: the background genotype (3610, 168) is not always mentioned (MB8 and DTU strains)

Line 369 - does strain P9-B1 has an SPbeta-like phage in the *spsM* locus?

Line 375-380 - I wrote this before reaching the end of the manuscript, but it reflects the vagueness of the description: "not sure I understand if these results are in contradiction with the previous work by Sanchez-Vizueté or are simply different because of strain background and the different method of "deleting" *spsM* (through actual deletion or through phage integration). This should be more clearly written. "

Line 411-418 - *swrA* has been shown to be selected for deletions in other works and is one of the domesticating mutations of the lab strain - this is mentioned in the discussion, but should be mentioned here. Is the mutation in the same strain identified by Kearns?

Lines 633-639 - the difference between the spontaneous *aimP* mutation and the *aimP::ery* may also stem from an effect on the expression of the downstream *comA* gene. Should be mentioned as another option.

Line 686 - what is the "Lys SPβ background" strain??

Reviewer #2 (Comments for the Author):

Here, I review the manuscript "Disruption of bacteriophage integration site promotes rapid diversification of multicellular traits in *Bacillus subtilis*" by Maja Popović and colleagues, submitted for publication in Microbiology Spectrum. All in all, the authors show that unexpected results are among the most exciting in biology and lead to new insights formerly not envisioned. The paper is generally well written section-by-section, but the way all the parts are interpreted and packaged and framed is problematic and needs adjustments. Below, I outline my main concerns about aspects of the manuscript that require scrutiny.

Comment 1: Problematic framing - general remarks.

The authors state in various places that they set out to study the ecological and evolutionary consequences of prophage integration into *spsM* in *B. subtilis*, an important locus implicated in biofilm production (see Introduction and Importance Statement). Yet, instead of delivering on this aspect, their essential observation is that different kinds of disruption of the *spsM* gene lead to different kinds of secondary mutations in different strains. From this observation, the authors conclude that prophage integration into *spsM* makes the genome more robust to new mutations (in contrast to the artificial integration of a kanamycin resistance cassette). Here, I beg to differ, as the author's conclusion lacks robust data. I offer several points below that hopefully help to clarify the situation.

Comment 2: Distinguishing the relative contributions of mutation and selection

The integration of a large prophage into an essential gene for biofilm-formation by definition lowers genome stability both with respect to the wildtype and to the *spsM*-mutant harbouring a smaller kanamycin-resistance cassette. As such, invoking higher genome stability in prophage-insertion backgrounds is not a very convincing argument. Instead, the authors should refer to more appropriate technical terms and explicitly refer to mutation rates and selection coefficients where appropriate. In more detail, the authors observed higher mutant frequencies on colony edges under stress and interpreted them as higher diversification rates (due to lower genomic stability). Here, they essentially assert that the genomes appear to accumulate more new mutations. Yet they do not offer any mutation rate estimates or explain how selection is shaping the outcome. In my mind, the observed high mutational parallelism in *spsM*-disruption mutants is unlikely due to the *spsM*-disruption itself. This is because most observed mutations (those in *swrA*) manifest in known mutational hotspots (microsatellites, poly-T-tracts), which are prone to elevated mutation rates. Thus, it is not necessary to think that *spsM*-insertions somehow change the underlying polymerase-strand-slippage rates even further at these microsatellite sites. Instead, it is more reasonable to assume that high parallelism is due to a combination of mutation and selection. A high number of mutants might arise due to both elevated local mutation supply rates (at certain hyper-mutable positions) and high population size (cell numbers), which both increase population-level mutation rates, and could be further boosted under favorable conditions (high selective advantage of outgrowth at colony edges that increases accessibility to free nutrients due to limited competition). This can more easily explain the frequent recurrent outgrowth of the same mutants than hyper-hypermutators. The authors do offer a variant of this alternative interpretation in their discussion and need to consider this as the more likely scenario than genome stability.

Comment 3: Quantifying the relative contributions of mutation and selection

The authors discuss that there is evidence for elevated divergence rates in certain genetic backgrounds, but they lack the necessary quantitative data to bolster their claims. Given the number of experiments and rather strong data underlying their work, I think they would be in a strong position to collect the missing data. In more detail, the authors would need to estimate three main parameters for any of their arguments:

- the mutation supply rate (μ), to estimate the chance for secondary mutations (in *swrA* and *comP*) to arise per cell.
- the population size (N), or cell number for each mutant background, to estimate the number of mutants (in *swrA* and *comP*) that arise per microcolony.
- The fitness consequences of both their various *spsM*-insertions and secondary mutations (*swrA* and *comP*) in their relevant genetic backgrounds relative to a common isogenic ancestor.

Mutation rates (μ) for different sites are known for some *B. subtilis* strains and could be drawn from the relevant papers (Sung et al., 2015; <https://doi.org/10.1093/molbev/msv055>).

As to fitness estimates, they offer several relative fitness measurements already, but they need to make sure that all mutants of the same genetic background are matched against the same ancestor to infer comparable fitness consequences of their mutations (see Wisner & Lenski, 2013; <https://doi.org/10.1126/science.1243357>).

Finally, for N , it could be sufficient to estimate cell numbers for a typical microcolony and link this number to the number of average outgrowths of a mutation type. From this, the authors could estimate the emergence rate of different mutants, and by considering differences in mutation rates and selection coefficients, work out divergence rates, and how strong the impact of selection versus mutation (or drift) might be to explain their observation.

Comment 4: The impact of epistasis

The authors seem to favour the genomic stability hypothesis via prophage integration, because different *spsM*-insertions

(prophage versus kan-resistance cassette) qualitatively differ in terms of visible secondary mutant outgrowths on colony edges. However, this could be a consequence of epistasis. Differently inserted elements likely entail different epistatic interactions with newly arising mutations under certain conditions and hence might result in different selection coefficients. This might also translate to visible differences in the number of outgrowing sectors at colony edges. Epistatic interactions also likely vary across completely different genetic backgrounds (NDMed vs. P9_B1, etc.). To explain the differences across genetic backgrounds, the authors should reveal evolutionary distances between strains and carefully discuss epistasis. As a second step, for a stronger quantitative argument, they would need to resort to their (new) fitness assays from above and assess if there is an indication for epistasis.

Comment 5: Phenotypes and phage adsorption

The authors discuss that the SP-beta phage may manipulate its own integration and excision into and from the host genome. This sounds plausible, but I think the authors need to better explain their logic, considering a few peculiarities in their system, which I explain below. For any integration to happen into *spsM*, the phage first must adsorb to the host cell and inject its DNA. It is known that increased mucoidy (sliminess) in bacteria such as *E. coli* prevents phage infection more effectively. So, a slimier host is typically harder to infect in the first place. Thinking of your system, why does phage SP-beta infect hosts that produce biofilms via ECM when non-infected? Wouldn't they prefer non-biofilm producers that are less difficult to attack? And of all places, why would they insert exactly into *spsM* that reduces ECM-production, when disrupted? In the latter case, hosts are even more accessible to other phages (i.e., competitors). In short, *spsM* is a highly interesting location, and hints at a link between biofilm production and phage infection, but, curiously, the phenotypic consequences of insertion and excision are exactly the opposite of what is expected from other bacterial systems and require careful explanation of who controls whom and why. As an aside, what happens after prophage excision? Usually, excision leads the phage to enter the lytic cycle and lyse the host cell. This part of the cycle needs to be better reflected in the discussion.

Typos, Interpunctuation:

I.145: Typo: "form -80{degree sign}C" should be "from -80{degree sign}C"

I. 231: Missing article: "in custom C++" should be "in a custom C++"

I. 239: the multiplication sign in the formula can be removed as in the original paper

I. 241: italicize for consistency: "<math>I(x)I(y)>"

II. 241- 242: this is a direct quote of the original paper, which is neither indicated by quotation marks nor by citation in brackets. I suggest paraphrasing this sentence.

REBUTTAL LETTER

We thank both reviewers for their time and helpful comments. We have revised the manuscript to address all points raised. Below we provide the **reviewer's comments in bold** and *our responses in italic*. Please see the **Popovic_et_al_revised_changes_highlighted.docx** file for details and for the exact lines where major revisions were made.

Reviewer #1 (Comments for the Author):

The manuscript by Popović et al studies the effect of either deletion of, or integration into, the *spsM* gene, where SPbeta phage integrates in the lab strain. To this aim, they mostly examine the effect of this deletion in two wild isolates. This is a follow-up paper to a previous work from the same lab (Sanchez-Vizueté et al., 2015) which identified changes in biofilm phenotype upon deletion of this gene in one of the wild isolates.

The work has several main results:

1. The previous work was based on an unfortunate artifact - the *spsM* (*ypqP*) mutant strain examined by Sanchez-Vizueté had an additional mutation in the *comP* gene. This mutation led to the biofilm phenotype led to the perturbed biofilm phenotype and not the *spsM* mutation. The author has previously done complementation assays, but apparently the *comP* mutation did not appear in the *spsM* complementation strain. Therefore, complementation seemed to indicate a direct role of *spsM* in biofilm formation.
2. The *spsM* mutant accumulated additional mutations at the edge of the colony in either the *comP* and *swrA* genes (and in an additional unidentified locus). These mutations tend to have a strong (and different) biofilm phenotype, which is carefully characterized by the authors. It is shown that these mutations are selected under strong biofilm producing conditions.
3. Mutations in *comP* and *swrA* tend to form at a much higher rate on the *spsM::kan* mutant than on the wild-type. This is still true in the *spsM* complementation strain. The SPbeta lysogen of one of the wild isolates showed some mutant phenotypes, but at a much lower rate.

General:

It is to be commended that the authors alert the community to an unintentional mistake which occurred in a previous manuscript. The analysis of the biofilm phenotype of the *comP* and *swrA* mutant is also interesting, though I think that it generally matches what's expected from the deletions based on what is known about their role in strain 3610 - the model strain for biofilm formation. However, I find the main claim of the manuscript - that *spsM* is an evolutionary hotspot that influence genome integrity and adaptive potential - to be unsatisfactory and over-hyped. The most likely hypothesis in my mind is that the deletion marker leads to over- or under-expression of a neighboring gene, which puts a

stronger selection pressure for compensating biofilm mutants. This is not really what one would call a hot spot for adaptive potential. Further work should be done to estimate the sources for the increased phenotypic diversification rate.

*We thank the reviewer for the constructive general assessment. We agree that our initial framing overstated the evolutionary implications of *spsM* disruption, and we have revised the manuscript accordingly. In the revised version, we no longer describe *spsM* as an evolutionary hotspot or invoke enhanced genome integrity/adaptive potential. Instead, we interpret the recurrent *swrA/comP* variants within a mutation–selection framework and emphasize that the elevated mutation rate is specific to the *spsM::kan* insertion rather than to loss of *spsM* function. Mechanistic possibilities, including cassette-linked effects on neighboring genes, are explicitly discussed and are addressed in detail in responses to Major points below.*

Major points:

1. The role of the *spsM::kan* mutation in supporting increased phenotypic (and genetic) diversity. The most likely source of the effect is the selection marker and its promoter. The authors should re-estimate the rate of diversification in a marker-less deletion of *spsM* - this is easy to do and would greatly help in distinguishing between a cis effect of the DNA sequence of the *spsM* locus and a trans effect of marker expression (most likely) on neighboring genes.

If the increased genetic diversity is specific to the kan-marked mutant, I would suggest studying the over expression or deletion (according to the presumed effect) of the genes neighbouring the *spsM* gene. In strain 168 these are the *msrAB* genes (Methionine sulfoxide reductases) or *ypoP* (a transcription factor of unknown function) on one side and the *phy* (phytase) gene or *cge* operon (spore maturation) on the other side (based on SubtiWiki) - both are not known to affect selection on biofilm, but they may have such an effect. Alternatively, these might be different genes in the wild isolates used by the authors.

If the authors find that the marker-less deletion still has the increased diversity effect, it probably suggests the presence of a regulatory region on that locus. I would therefore also recommend doing RT-PCR of the neighboring genes in different genetic backgrounds to verify an effect on their expression level.

*We appreciate the reviewer's suggestions and are thankful for this insightful comment. To directly test whether the diversification phenotype reflects loss of *spsM* or effects of the *kan* cassette, we constructed a markerless Δ *spsM* mutant as suggested (see Methods lines 228-241) and quantified mutation rates using a rifampicin fluctuation assay (see Methods lines 388-411, see Results lines 787-814). The *spsM::kan* strain showed a significantly elevated mutation rate compared to wild type, whereas the mutation rate of markerless Δ *spsM* did*

not statistically differ from wild type and was statistically lower compared to *spsM::kan*. The *SPβ* lysogen also did not show a statistically significant increase, and complementation of *spsM* at *amyE* did not fully restore the *spsM::kan* mutation rate to the WT level. Results are presented as updated Figure 7 (see Results lines 787-814) (see figure below).

Figure 7: Mutation rate estimates for *B. subtilis* P9_B1 derivatives. Mutation rates are estimated from rifampicin fluctuation assays. Columns represent mean mutation rates ($n = 84$) and error bars indicate standard deviation. Mutation parameters were determined using RStudio *flan* package. Final population sizes measured for each culture were used to convert mutation parameters to per-cell mutation rates. Pairwise comparisons between strains were performed using *flan*'s two-sample likelihood ratio test. Statistical significance is indicated as $*p < 0.05$, $**p < 0.005$, $***p < 0.001$.

In addition, CFU measurements from macrocolonies and planktonic cultures indicate comparable population sizes (N) across backgrounds, supporting increased mutation supply (rather than higher N) as the driver of the higher visible-variant frequency in *spsM::kan* (see supplementary figure S6 (see figure below), see Results lines 780-785). Accordingly, we no longer interpret *spsM* as an intrinsic hotspot. Instead, we attribute the effect to the specific *spsM::kan* insertion (most consistent with a cis/polar effect at the locus) (see Results lines 789-814). We now explicitly discuss candidate neighboring genes (*msrAB* downstream; *phyC/cgeAB* upstream) and note that resolving the transcriptional mechanism would require targeted expression measurements (e.g., RT-qPCR/RT-PCR) (see Discussion lines 879-905), which we did not pursue here because it is beyond the scope of this study.

Figure S6: Population size measurements of *B. subtilis* P9_B1 derivatives. CFU/ml values of macrocolonies and planktonic cultures for P9_B1 wt, *spsM::kan*, *spsM::kan amyE::spsM*, SPβ lysogen and Δ *spsM* strains. Macrocolonies were grown on LB agar at 30 °C for 48 hours prior to CFU determination, whereas planktonic cultures represent the values acquired during the fluctuation assay. Columns show mean values (macrocolony: n=3; planktonic: n=6) and error bars indicate standard deviation. Statistical analysis was performed using one-way ANOVA with Tukey's Honest Significant Difference (HSD) test. Different letters above bars denote statistically significant differences between strains ($p < 0.05$)

2. The claim of increased phenotypic diversity in mutants and lysogen, compared to wild-type should be better quantified. Specifically, the claim that the lysogen led to the emergence of mutant in one out of 40 colonies while no mutants were observed in the wild-type is definitely not statistically significant. It is easy to estimate the rate of emergence of visible mutant phenotype in several hundreds of colonies of each type and do a proper statistical analysis.

We appreciate this point and agree that our initial phrasing overstated what could be concluded from limited screening. In the revised Results, we clarify the screening design and report the quantitative outgrowth counts from four independent experiments (140 macrocolonies per strain) (see Results lines 738-745), with per-experiment counts provided transparently in Supplementary Figure S5. We do not claim statistical significance for the single visible variant observed in the SPβ lysogen; rather, we describe it as a rare event under our conditions.

*To provide a rigorous quantitative basis for differences in diversification, we added rifampicin fluctuation assays showing that *spsM::kan* significantly elevates mutation rate relative to wild type, whereas the SPβ lysogen does not show a statistically significant increase (see Results lines 787-814). Together, these additions support the interpretation that the higher frequency of visible variants in *spsM::kan* is consistent with increased mutation supply, while avoiding over-interpretation of the lysogen screen.*

3. The paper's writing is somewhat convoluted. I would have liked to understand that the previously associated biofilm phenotype of the *spsM* mutant was a mistake before reaching line 535. I think that the claims on being a "hot-spot" for diversity should be

toned down even if they find a different mechanism underlying this process than what I suspect it is.

*We acknowledge the reviewer's concern and thank the reviewer for noting that the correction regarding the previously reported NDmed *spsM::kan* biofilm phenotype appeared too late. In the revised manuscript, we state this explicitly already in the Abstract (see Abstract lines 22–27). We also rewrote the beginning of the Results to provide a clearer setup of the study and the key initial observations (see Results lines 466-473): we first restate the original expectation based on prior work, then immediately show that we can reproduce the reported flat NDmed *spsM::kan* (GM3248) phenotype, and then show that introducing the identical *spsM::kan* allele into P9_B1 does not yield the same morphology, and that SPβ lysogeny likewise does not recapitulate the flat phenotype. To help guide the reader, we added an explicit transition sentence ("The contrasting effects of the same *spsM::kan* mutation in NDmed and P9_B1 are examined in more detail in the following sections." (see Results lines 499-505)), signaling that the apparent discrepancy is addressed later in the Results. In parallel, we toned down the overall framing by removing "hotspot/genome stability" language and focusing instead on the mutation-rate data and mutation–selection dynamics underlying recurrent *swrA* and *comP* variants under our laboratory conditions (see Discussion lines 922-937).*

Minor:

Line 189: the background genotype (3610, 168) is not always mentioned (MB8 and DTU strains)

We thank the reviewer for pointing out this discrepancy. MB8_B7 and PS-216 are genetic backgrounds (labels have now also been added to column "Background and genotype" for clarity) (see Table 1 in Methods lines 243-244). DTUB strains have P9_B1 genetic background and have been moved to the P9_B1 section of the table (see Table 1 in Methods lines 243-244).

Line 369 - does strain P9-B1 has an SPbeta-like phage in the *spsM* locus?

*P9_B1 does not carry an SPβ-like prophage at *spsM*. We now clarify this at the mentioned point in the text, for higher clarity (see Results lines 495-496).*

Line 375-380 - I wrote this before reaching the end of the manuscript, but it reflects the vagueness of the description: "not sure I understand if these results are in contradiction with the previous work by Sanchez-Vizueté or are simply different because of strain background and the different method of "deleting" *spsM* (through actual deletion or through phage integration). This should be more clearly written. "

We thank the reviewer for highlighting this potential confusion and have revised the text wording for higher clarity. In the revised manuscript, we rewrote this part of the Results to make the results and conclusion clearer (see Results lines 489–505). Specifically, we now state unambiguously that the *spsM::kan* allele introduced into P9_B1 is the same construct as in NDmed *spsM::kan* (GM3248), so the differing colony morphologies cannot be attributed to different disruption designs but instead reflect background-dependent outcomes. We also explicitly separate the cassette-based disruption from SP β lysogeny by stating that neither NDmed nor P9_B1 SP β lysogens show the flat NDmed *spsM::kan* phenotype (see Results lines 502-503), and we include a short transition sentence directing readers to later Results sections where the NDmed phenotype is mechanistically resolved (see Results lines 499-500).

Line 411-418 - *swrA* has been shown to be selected for deletions in other works and is one of the domesticating mutations of the lab strain - this is mentioned in the discussion, but should be mentioned here. Is the mutation in the same strict identified by Kearns?

We thank the reviewer for this helpful suggestion. In the revised manuscript we now clarify that the spontaneous *swrA* mutation observed in the P9_B1 fluffy strain occurs within the same homopolymeric T tract previously identified as a domestication hotspot in laboratory strains (see Results lines 531-537). Our sequencing shows that the wild-type allele contains eight T's, the P9_B1 fluffy mutant carries a seven-T allele (c.26delT), and *B. subtilis* 168 carries a nine-T allele. Thus, although the specific indel differs, all mutations arise at the same tract described by Kearns and colleagues (2004) as prone to slipped-strand mispairing and frameshift formation. This information has been added to the Results section as requested (see Results lines 531-537).

Lines 633-639 - the difference between the spontaneous *aimP* mutation and the *aimP:ery* may also stem from an effect on the expression of the downstream *comA* gene. Should be mentioned as another option.

We appreciate this note. We agree that altered expression of the downstream gene *comA* represents a plausible alternative explanation for the phenotypic differences between the two *comP* alleles. We have now incorporated this possibility into the Results section (see Results lines 719-721).

Line 686 - what is the "Lys SP β background" strain??

The term "Lys SP β background" was intended to refer to the SP β -lysogenized P9_B1 strain. To avoid ambiguity, we have revised the sentence to contain "the P9_B1 Lys SP β strain," which clearly identifies the genotype and resolves the misunderstanding (see Results line 726).

Reviewer #2 (Comments for the Author):

Here, I review the manuscript "Disruption of bacteriophage integration site promotes rapid diversification of multicellular traits in *Bacillus subtilis*" by Maja Popović and colleagues, submitted for publication in *Microbiology Spectrum*. All in all, the authors show that unexpected results are among the most exciting in biology and lead to new insights formerly not envisioned. The paper is generally well written section-by-section, but the way all the parts are interpreted and packaged and framed is problematic and needs adjustments. Below, I outline my main concerns about aspects of the manuscript that require scrutiny.

We appreciate the reviewer's positive assessment of the experimental work and the emphasis on the value of the unexpected findings that motivated this study. We also recognize the concern regarding framing. Accordingly, we substantially revised the manuscript to clarify interpretations, avoid overstatements, and align the narrative with the quantitative data now included (discussed in detail below). We now frame the results explicitly in terms of mutation supply and selection rather than broad claims about "genome stability" or evolutionary potential (discussed in detail below).

Comment 1: Problematic framing - general remarks.

The authors state in various places that they set out to study the ecological and evolutionary consequences of prophage integration into *spsM* in *B. subtilis*, an important locus implicated in biofilm production (see Introduction and Importance Statement). Yet, instead of delivering on this aspect, their essential observation is that different kinds of disruption of the *spsM* gene lead to different kinds of secondary mutations in different strains. From this observation, the authors conclude that prophage integration into *spsM* makes the genome more robust to new mutations (in contrast to the artificial integration of a kanamycin resistance cassette). Here, I beg to differ, as the author's conclusion lacks robust data. I offer several points below that hopefully help to clarify the situation.

*We appreciate the reviewer's concern that our original framing overreached and was not sufficiently supported by quantitative evidence. In the revised manuscript, we removed claims that SPβ integration at *spsM* increases "genome robustness" and instead anchored our interpretation in direct mutation-rate measurements and population-size controls. Specifically, we now performed and include rifampicin fluctuation-based mutation-rate estimates (see Methods lines 388-411, see Results lines 787-814) that show P9_B1 *spsM::kan* has a significantly elevated per-cell mutation rate relative to wild type (2.15×10^{-9} vs 4.90×10^{-10} ; $p = 0.00316$), whereas a newly prepared markerless Δ *spsM* strain does not differ from wild type (8.03×10^{-10} vs 4.90×10^{-10} ; $p = 0.42790$) and remains significantly lower than *spsM::kan* ($p = 0.02188$). Results are presented as updated Figure 7 (see Results lines 787-814) (see figure below).*

Figure 7: Mutation rate estimates for *B. subtilis* P9_B1 derivatives. Mutation rates are estimated from rifampicin fluctuation assays. Columns represent mean mutation rates ($n = 84$) and error bars indicate standard deviation. Mutation parameters were determined using RStudio *flan* package. Final population sizes measured for each culture were used to convert mutation parameters to per-cell mutation rates. Pairwise comparisons between strains were performed using *flan*'s two-sample likelihood ratio test. Statistical significance is indicated as $*p < 0.05$, $**p < 0.005$, $***p < 0.001$.

Together with the explicit reporting of outgrowth counts across independent experiments and the competition data for *swrA* loss under LBGM macrocolony conditions, the revised manuscript reframes our conclusions around mutation supply and selection (see Discussion lines 968-983) and does not infer adaptive phage-mediated manipulation of biofilm via *spsM* from our data.

Comment 2: Distinguishing the relative contributions of mutation and selection

The integration of a large prophage into an essential gene for biofilm-formation by definition lowers genome stability both with respect to the wildtype and to the *spsM*-mutant harbouring a smaller kanamycin-resistance cassette. As such, invoking higher genome stability in prophage-insertion backgrounds is not a very convincing argument. Instead, the authors should refer to more appropriate technical terms and explicitly refer to mutation rates and selection coefficients where appropriate. In more detail, the authors observed higher mutant frequencies on colony edges under stress and interpreted them as higher diversification rates (due to lower genomic stability). Here, they essentially assert that the genomes appear to accumulate more new mutations. Yet they do not offer any mutation rate estimates or explain how selection is shaping the outcome. In my mind, the observed high mutational parallelism in *spsM*-disruption mutants is unlikely due to the *spsM*-disruption itself. This is because most observed mutations (those in *swrA*) manifest in known mutational hotspots (microsatellites, poly-T-tracts), which are prone to elevated

mutation rates. Thus, it is not necessary to think that *spsM*-insertions somehow change the underlying polymerase-strand-slippage rates even further at these microsatellite sites. Instead, it is more reasonable to assume that high parallelism is due to a combination of mutation and selection. A high number of mutants might arise due to both elevated local mutation supply rates (at certain hyper-mutable positions) and high population size (cell numbers), which both increase population-level mutation rates, and could be further boosted under favorable conditions (high selective advantage of outgrowth at colony edges that increases accessibility to free nutrients due to limited competition). This can more easily explain the frequent recurrent outgrowth of the same mutants than hyper-hypermutators. The authors do offer a variant of this alternative interpretation in their discussion and need to consider this as the more likely scenario than genome stability.

The reviewer is correct that our original “genome stability/robustness” framing was not technically appropriate. In the revised manuscript we removed that language and rewrote the Results/Discussion to treat visible edge outgrowth frequency as an outcome of mutation supply and selection, not as a direct readout of “genome integrity” (see Results lines 780-814, see Discussion lines 867-877)

To address the specific concern that we previously lacked mutation-rate estimates, we added a rifampicin fluctuation assay to quantify per-cell mutation rates across relevant P9_B1 genotypes (see Results lines 887-814). This shows that *spsM::kan* has a significantly higher mutation rate than wild type (2.15×10^{-9} vs 4.90×10^{-10} ; $p = 0.00316$), while the *SP8* lysogen does not show a statistically significant increase relative to wild type (1.08×10^{-9} ; $p = 0.07621$). We also added CFU measurements from macrocolonies and from planktonic cultures used in the fluctuation assay (see Results lines 780-785 and lines 794-795), which indicate comparable population sizes across backgrounds, arguing against a simple “larger *N*” explanation for higher outgrowth frequency in *spsM::kan* (see supplementary figure S6 (see figure below)).

Figure S6: Population size measurements of *B. subtilis* P9_B1 derivatives. CFU/ml values of macrocolonies and planktonic cultures for P9_B1 wt, *spsM::kan*, *spsM::kan amyE::spsM*, *SP8* lysogen and Δ *spsM* strains. Macrocolonies were grown on LB agar at 30 °C for 48 hours prior

to CFU determination, whereas planktonic cultures represent the values acquired during the fluctuation assay. Columns show mean values (macrocolony: $n=3$; planktonic: $n=6$) and error bars indicate standard deviation. Statistical analysis was performed using one-way ANOVA with Tukey's Honest Significant Difference (HSD) test. Different letters above bars denote statistically significant differences between strains ($p < 0.05$).

Accordingly, we now interpret the repeated emergence of *swrA/comP*-associated morphotypes under macrocolony conditions as consistent with increased mutation supply in *spsM::kan* coupled to strong spatial selection at the expanding colony edge (see Discussion lines 922-937), without invoking a prophage-associated "stability" effect.

Comment 3: Quantifying the relative contributions of mutation and selection

The authors discuss that there is evidence for elevated divergence rates in certain genetic backgrounds, but they lack the necessary quantitative data to bolster their claims. Given the number of experiments and rather strong data underlying their work, I think they would be in a strong position to collect the missing data. In more detail, the authors would need to estimate three main parameters for any of their arguments:

- the mutation supply rate (μ), to estimate the chance for secondary mutations (in *swrA* and *comP*) to arise per cell.
- the population size (N), or cell number for each mutant background, to estimate the number of mutants (in *swrA* and *comP*) that arise per microcolony.
- The fitness consequences of both their various *spsM*-insertions and secondary mutations (*swrA* and *comP*) in their relevant genetic backgrounds relative to a common isogenic ancestor.

Mutation rates (μ) for different sites are known for some *B. subtilis* strains and could be drawn from the relevant papers (Sung et al., 2015; <https://doi.org/10.1093/molbev/msv055>).

As to fitness estimates, they offer several relative fitness measurements already, but they need to make sure that all mutants of the same genetic background are matched against the same ancestor to infer comparable fitness consequences of their mutations (see Wisner & Lenski, 2013; <https://doi.org/10.1126/science.1243357>).

Finally, for N , it could be sufficient to estimate cell numbers for a typical microcolony and link this number to the number of average outgrowths of a mutation type. From this, the authors could estimate the emergence rate of different mutants, and by considering differences in mutation rates and selection coefficients, work out divergence rates, and how strong the impact of selection versus mutation (or drift) might be to explain their observation.

We thank the reviewer for this comment and address it as follows. In the revised manuscript, we addressed the reviewer's request by adding new measurements (with mutation-rate framing and population-size controls already described in our responses to Comments 1–2).

*Mutation supply (μ): We added rifampicin fluctuation assays (see Results lines 787-814) with per-cell mutation-rate estimates (via fln using measured final population sizes) and included a markerless Δ spsM strain to distinguish effects of SpsM loss from effects of the *spsM::kan* allele. These data show that the elevated mutation rate is specific to the cassette insertion background rather than *spsM* inactivation per se. This is now discussed in detail in the rewritten discussion (see Discussion lines 922-937)*

*

Population size (N): We added CFU quantification from macrocolonies (see Results lines 780-785) and from the planktonic cultures used for mutation-rate determination (see Results lines 794-795), showing comparable final cell numbers across the key genotypes. This supports interpreting differences in visible outgrowth frequency as differences in mutation supply and/or selection, rather than systematically larger populations.

*Fitness consequences: We already showed competition assays under biofilm-promoting macrocolony conditions (LBGM) showing that loss of *swrA* can confer a reproducible competitive advantage in the *spsM::kan* background, consistent with selection favoring reduced investment in costly social behaviors under laboratory conditions (see Discussion lines 968-974). We also note explicitly that fitness effects of *comP* variants and of different *spsM* disruption modes were not systematically quantified (see Discussion lines 973-974).*

*Finally, prompted by the reviewer's suggestion, we checked Sung et al. (2015) and added text noting that genome-wide mutation-accumulation analyses do not single out *swrA* or *comP* as exceptional hypermutable regions (see Discussion lines 979-983). We therefore interpret their repeated recovery in our macrocolonies as primarily reflecting selection on accessible, high-effect regulatory targets (with *swrA* additionally containing a simple-sequence tract prone to recurrent indels), rather than requiring a special local "hypermutability" explanation.*

Comment 4: The impact of epistasis

The authors seem to favour the genomic stability hypothesis via prophage integration, because different *spsM*-insertions (prophage versus *kan*-resistance cassette) qualitatively differ in terms of visible secondary mutant outgrowths on colony edges. However, this could be a consequence of epistasis. Differently inserted elements likely entail different epistatic interactions with newly arising mutations under certain conditions and hence might result in different selection coefficients. This might also translate to visible differences in the number of outgrowing sectors at colony edges. Epistatic interactions also likely vary across completely different genetic backgrounds (NDMed vs. P9_B1, etc.). To explain the differences across genetic backgrounds, the authors should reveal evolutionary distances between strains and carefully discuss epistasis. As a second step,

for a stronger quantitative argument, they would need to resort to their (new) fitness assays from above and assess if there is an indication for epistasis.

*We thank the reviewer for raising the possibility that our observations could reflect epistatic interactions and for requesting information on evolutionary distances between the strain backgrounds. In the revised manuscript we now explicitly address this point by adding a phylogenetic analysis and ANI estimates comparing P9_B1 and NDmed (see Supplementary Figure S7). Using autoMLST2 and whole genome sequences, we show that P9_B1 and NDmed cluster closely together among other *B. subtilis* strains (see new Supplementary Figure S7 (see below)), and ANI between the two genomes is ~99.96%. These data indicate that the diversification patterns we report arise in broadly similar genomic backgrounds, rather than in highly divergent lineages.*

Figure S7: Phylogenetic tree showing genetic similarity between *B. subtilis* P9_B1 (red) and *B. subtilis* NDmed (blue) compared to other closely related strains. The tree was obtained using autoMLST2 and visualized in iTOL. Branch lengths represent substitutions per nucleotide site. The tree scale (0.1) refers to the units of branch length displayed below the tree.

*At the same time, we agree that epistasis remains a plausible contributor to the observed differences between distinct forms of *spsM* disruption. In the revised Results, we therefore interpret our data within a mutation–selection framework, supported by the new fluctuation assay and CFU measurements, and we explicitly state that differences between the kanamycin-marked *spsM* allele and *SPB* integration may reflect local, background specific interactions between these *spsM* disruptions and secondary regulatory mutations such as*

those in *swrA* and *comP*. We develop this point in the revised Discussion to evaluate potential epistatic effects more explicitly (see Discussion lines 986-990).

Comment 5: Phenotypes and phage adsorption

The authors discuss that the SP-beta phage may manipulate its own integration and excision into and from the host genome. This sounds plausible, but I think the authors need to better explain their logic, considering a few peculiarities in their system, which I explain below. For any integration to happen into *spsM*, the phage first must adsorb to the host cell and inject its DNA. It is known that increased mucoidy (sliminess) in bacteria such as *E. coli* prevents phage infection more effectively. So, a slimier host is typically harder to infect in the first place. Thinking of your system, why does phage SP-beta infect hosts that produce biofilms via ECM when non-infected? Wouldn't they prefer non-biofilm producers that are less difficult to attack? And of all places, why would they insert exactly into *spsM* that reduces ECM-production, when disrupted? In the latter case, hosts are even more accessible to other phages (i.e., competitors). In short, *spsM* is a highly interesting location, and hints at a link between biofilm production and phage infection, but, curiously, the phenotypic consequences of insertion and excision are exactly the opposite of what is expected from other bacterial systems and require careful explanation of who controls whom and why. As an aside, what happens after prophage excision? Usually, excision leads the phage to enter the lytic cycle and lyse the host cell. This part of the cycle needs to be better reflected in the discussion.

*We appreciate this thoughtful comment and agree that our original framing might have been unclear. Importantly, our new data directly contradict the earlier model in which SPβ would manipulate host biofilm architecture via *spsM*. We now show that loss of *spsM* alone does not reproducibly alter biofilm morphology, and that previously reported effects were due to secondary mutations. Thus, our revised manuscript no longer proposes that SPβ integrates into *spsM* to modulate ECM or biofilm properties (see Discussion lines 1056-1060). We also clarify that SPβ excises primarily during sporulation, remains episomal and is carried into spores, while spontaneous excision leading to lysis appears rare (see Discussion lines 916-920).*

*Using reconstructed and control backgrounds, we show that *spsM* disruption alone does not reproducibly impair biofilm formation and that the previously reported NDmed *spsM::kan* biofilm defect instead derives from an undetected secondary *comP* mutation (see Results 665-686). Consistent with this, our fluctuation assay does not detect a statistically significant increase in mutation rate in the SPβ lysogen relative to wild type under our assay conditions (see Results lines 787-814), and we therefore do not infer ecological “who-controls-whom” scenarios (e.g., adsorption/biofilm accessibility tradeoffs) from these data. We also clarified the life-cycle interpretation by distinguishing precise developmental excision in the active-lysogeny context (restoring *spsM* during sporulation and allowing episomal maintenance)*

from mitomycin C–induced prophage induction, which represents DNA-damage–triggered lytic entry and is not equivalent to developmental excision (see Discussion lines 915-920).

Typos, Interpunctuation:

I.145: Typo: "form -80{degree sign}C" should be "from -80{degree sign}C"

Has been corrected.

I. 231: Missing article: "in custom C++" should be "in a custom C++"

Has been corrected.

I. 239: the multiplication sign in the formula can be removed as in the original paper

Has been corrected.

I. 241: italicize for consistency: "< I(x)I(y)>"

Has been corrected.

II. 241- 242: this is a direct quote of the original paper, which is neither indicated by quotation marks nor by citation in brackets. I suggest paraphrasing this sentence.

The sentence has been changed, to not be a direct quotation.

Re: Spectrum02504-25R1 (Disruption of bacteriophage integration site promotes rapid diversification of multicellular traits in *Bacillus subtilis*)

Dear Dr. Anna Dragoš:

Thank you for resubmitting your work to mSpectrum. Below you will find my comments, instructions from the Spectrum editorial office, and the reviewers comments. They are both very positive and agree that you have done an excellent work in responding to their previous comments. You will see below very minor and quick changes are recommendd. I am thus happy to inform you that your manuscript is, in principle, accepted, pending these few modifications

Revision Guidelines

Sincerely,
Olaya Rendueles
Editor
Microbiology Spectrum

Reviewer #1 (Comments for the Author):

After reading the modified version I am more satisfied with the structure of the manuscript, with its interpretation of the data and with the results.

The authors show now that the increased frequency of mutants is attributed specifically for the kan insertion mutant, as I

suspected, and suggest this is most likely due to a polar impact on nearby genes. The authors show now an increase in mutational frequency (in an antibiotic resistance gene), to suggest that the effect of the kan insertion is on the number of mutations.

The data seem to support that interpretation, but does not rule out a parallel effect of the kan insert also on the strength of selection. All I ask is that this possibility would also be mentioned.

Other than that, I have no objection to the publication of the manuscript.

Reviewer #2 (Comments for the Author):

Here, I re-assessed a revised draft of "Disruption of bacteriophage integration site promotes rapid diversification of multicellular traits in *Bacillus subtilis*" by Maja Popović and colleagues.

I am very pleased that the authors took my concerns very seriously, and diligently revised their original manuscript accordingly. This included generating new data from fluctuation tests, phylogenetic tree inference and statistical analyses, beside other things.

I have no new additional requests, just a few typographical suggestions (See way down below, after my brief response on how well the authors handled my earlier comments here below).

Comment 1: Problematic framing - general remarks.

> The authors were very careful with their phrasing for the new draft. I think this version is on point in reflecting the observation and interpretation.

Comment 2: Distinguishing the relative contributions of mutation and selection

>The authors now offer mutation rate estimation from an appropriate fluctuation test for the different backgrounds. This is a great addition and strength of this new MS version, and I also commend the authors for using an appropriate tool and statistical evaluation. The addition of a new clean *spsM* deletion mutant demonstrates that mutation rates were significantly different in the *spsM::kan*-cassette insertion mutant. Very nice work.

Comment 3: Quantifying the relative contributions of mutation and selection

> Thanks for tackling this and teasing apart the various relative contributions. Again solves my concerns.

Comment 4: The impact of epistasis

>The phylogeny really adds to the quality of the paper, shows how the different strains are related, and thereby helps the reader evaluate genetic distances.

Comment 5: Phenotypes and phage adsorption

>In summary the authors come to the appropriate conclusion that the "new data directly contradict the earlier model in which SPβ would manipulate host biofilm architecture via *spsM*." and I agree that the data does not "support an interpretation in which SPβ integration at *spsM* adaptively manipulates biofilm formation. Any ecological scenarios linking *spsM* targeting to transmission remain speculative." (ll.915-920)

Typographical suggestions:

* Abstract: "Consistently [add comma here] we show that *spsM::kan* significantly elevates mutation rates,"

I guess this is alright: you just never cite Fig. 8B in the text, but you do reference in that order:

* Figure 8; Figure 8A; Figure 8C.

Maybe you want to call out Fig. 8B content here, too?

REBUTTAL LETTER

Please find the reviewers comments in **BOLD** and our responses in italics, directly below the comments.

Reviewer #1:

After reading the modified version I am more satisfied with the structure of the manuscript, with its interpretation of the data and with the results. The authors show now that the increased frequency of mutants is attributed specifically for the kan insertion mutant, as I suspected, and suggest this is most likely due to a polar impact on nearby genes. The authors show now an increase in mutational frequency (in an antibiotic resistance gene), to suggest that the effect of the kan insertion is on the number of mutations.

The data seem to support that interpretation, but does not rule out a parallel effect of the kan insert also on the strength of selection. All I ask is that this possibility would also be mentioned.

Other than that, I have no objection to the publication of the manuscript.

*We thank the reviewer for the thoughtful and constructive comments. As suggested, in the revised manuscript we have now explicitly acknowledged that although our data most strongly support an increased mutation supply in the *spsM::kan* background, a parallel effect on the strength of selection cannot be excluded. To address this point, we added the following clarification to the Discussion section:*

*“Although our data indicate that the elevated frequency of rifampicin resistant colonies in the *spsM::kan* background is most consistent with an increased mutation supply, we acknowledge that a parallel effect on the strength of selection cannot be ruled out. Antibiotic resistance cassettes can exert polar or off target effects on neighbouring genes, and such regulatory perturbations can modify antibiotic susceptibility phenotypes, thereby influencing the effective selection environment, as shown for cassette associated repression and downstream transcriptional changes in other bacterial systems (Mateus et al., 2021; Powers et al., 2022).”*

We are grateful for the positive assessment and for recommending publication. We sincerely thank the reviewer for the careful and constructive guidance throughout the revision process, which has strengthened the manuscript.

Reviewer #2:

Here, I re-assessed a revised draft of "Disruption of bacteriophage integration site promotes rapid diversification of multicellular traits in *Bacillus subtilis*" by Maja Popović and colleagues.

I am very pleased that the authors took my concerns very seriously, and diligently revised their original manuscript accordingly. This included generating new data from fluctuation tests, phylogenetic tree inference and statistical re-analyses, beside other things.

I have no new additional requests, just a few typographical suggestions (See way down below, after my brief response on how well the authors handled my earlier comments here below).

Comment 1: Problematic framing - general remarks.

> The authors were very careful with their phrasing for the new draft. I think this version is on point in reflecting the observation and interpretation.

Comment 2: Distinguishing the relative contributions of mutation and selection

>The authors now offer mutation rate estimation from an appropriate fluctuation test for the different backgrounds. This is a great addition and strength of this new MS version, and I also commend the authors for using an appropriate tool and statistical evaluation. The addition of a new clean *spsM* deletion mutant demonstrates that mutation rates were significantly different in the *spsM::kan-cassette* insertion mutant. Very nice work.

Comment 3: Quantifying the relative contributions of mutation and selection

> Thanks for tackling this and teasing apart the various relative contributions. Again solves my concerns.

Comment 4: The impact of epistasis

>The phylogeny really adds to the quality of the paper, shows how the different strains are related, and thereby helps the reader evaluate genetic distances.

Comment 5: Phenotypes and phage adsorption

>In summary the authors come to the appropriate conclusion that the "new data directly contradict the earlier model in which SP β would manipulate host biofilm architecture via *spsM*." and I agree that the data does not "support an interpretation in which SP β integration at *spsM* adaptively manipulates biofilm formation. Any ecological scenarios linking *spsM* targeting to transmission remain speculative." (ll.915-920)

We sincerely thank the reviewer for the very positive and encouraging assessment of our revised manuscript. We greatly appreciate the acknowledgment of the additional analyses we performed, including the new fluctuation tests, phylogenetic inference, and statistical re-analyses. We are pleased that these additions satisfactorily addressed the earlier concerns.

Typographical suggestions:

*** Abstract: "Consistently [add comma here] we show that *spsM::kan* significantly elevates mutation rates,"**

Done.

I guess this is alright: you just never cite Fig. 8B in the text, but you do reference in that order:

*** Figure 8; Figure 8A; Figure 8C.**

Maybe you want to call out Fig. 8B content here, too?

Implemented.

Re: Spectrum02504-25R2 (Disruption of bacteriophage integration site promotes rapid diversification of multicellular traits in *Bacillus subtilis*)

Dear Dr. Anna Dragoš:

Your manuscript has been accepted, and I am forwarding it to the ASM production staff for publication. Your paper will first be checked to make sure all elements meet the technical requirements. ASM staff will contact you if anything needs to be revised before copyediting and production can begin. Otherwise, you will be notified when your proofs are ready to be viewed.

Sincerely,
Olaya Rendueles
Editor
Microbiology Spectrum